# Matrix Wave^TM^ System for Mandibulo-Maxillary Fixation—Just Another Variation on the MMF Theme?—Part II: In Context to Self-Made Hybrid Erich Arch Bars and Commercial Hybrid MMF Systems—Literature Review and Analysis of Design Features

**DOI:** 10.3390/cmtr18030033

**Published:** 2025-07-15

**Authors:** Carl-Peter Cornelius, Paris Georgios Liokatis, Timothy Doerr, Damir Matic, Stefano Fusetti, Michael Rasse, Nils Claudius Gellrich, Max Heiland, Warren Schubert, Daniel Buchbinder

**Affiliations:** 1Department of Oral and Maxillofacial Surgery, Facial Plastic Surgery, Ludwig-Maximilians University Munich, 80337 Munich, Germany; paris.liokatis@med.uni-muenchen.de; 2Department of Otolaryngology, University of Rochester Medical Center, Rochester, NY 14642, USA; timothy_doerr@urmc.rochester.edu; 3London Plastic Surgery Centre, 123 Dundas St, London, ON N6A 1E8, Canada; dmatic@londonplasticsurgery.ca; 4Department of Maxillofacial Surgery, University of Padova Medical Center, 35122 Padova, Italy; stefano.fusetti@unipd.it; 5Department of Oral and Maxillofacial Surgery, Center of Reconstructive Surgery, University Hospital, Paracelsus Medical University, 5020 Salzburg, Austria; michael.rasse@outlook.at; 6Department of Oral and Maxillofacial Surgery, Hannover Medical School, 30625 Hannover, Germany; gellrich.nils-claudius@mh-hannover.de; 7Charité—Universitätsmedizin Berlin, Corporate Member of Freie Universität Berlin and Humboldt-Universität zu Berlin, 13353 Berlin, Germany; max.heiland@charite.de; 8Division of Plastic Surgery, University of Minnesota, Minneapolis, MN 55455, USA; warrenschubert@aol.com; 9Division of Oral and Maxillofacial Surgery, Department of Otolaryngology, Icahn School of Medicine at Mount Sinai, Mount Sinai Beth Israel, New York, NY 10003, USA; daniel.buchbinder@mountsinai.org

**Keywords:** mandibulo-maxillary fixation (MMF), bone anchorage, hybrid arch bars, hybrid MMF systems, design features, functionality, bone anchor holes, transalveolar screw fixation, tooth root injuries, interradicular targeting

## Abstract

Study design: Trends in the utilization of Mandibulo-Maxillary Fixation (MMF) are shifting nowadays from tooth-borne devices over specialized screws to hybrid MMF devices. Hybrid MMF devices come in self-made Erich arch bar modifications and commercial hybrid MMF systems (CHMMFSs). Objective: We survey the available technical/clinical data. Hypothetically, the risk of tooth root damage by transalveolar screws is diminished by a targeting function of the screw holes/slots. Methods: We utilize a literature review and graphic displays to disclose parallels and dissimilarities in design and functionality with an in-depth look at the targeting properties. Results: Self-made hybrid arch bars have limitations to meet low-risk interradicular screw insertion sites. Technical/clinical information on CHMMFSs is unevenly distributed in favor of the SMARTLock System: positive outcome variables are increased speed of application/removal, the possibility to eliminate wiring and stick injuries and screw fixation with standoff of the embodiment along the attached gingiva. Inferred from the SMARTLock System, all four CHMMFs possess potential to effectively prevent tooth root injuries but are subject to their design features and targeting with the screw-receiving holes. The height profile and geometry shape of a CHMMFS may restrict three-dimensional spatial orientation and reach during placement. To bridge between interradicular spaces and tooth equators, where hooks or tie-up-cleats for intermaxillary cerclages should be ideally positioned under biomechanical aspects, can be problematic. The movability of their screw-receiving holes according to all six degrees of freedom differs. Conclusion: CHMMFSs allow simple immobilization of facial fractures involving dental occlusion. The performance in avoiding tooth root damage is a matter of design subtleties.

## 1. Introduction

The technical aspects and applications of the Matrix Wave^TM^ MMF System (DePuy-Synthes) have been extensively detailed in a recent report (Cornelius et al. 2025, Part I) [1].

Further insights into the design and versatility of the Matrix Wave System are evident in the context of a comprehensive review of hybrid or bimodal MMF appliances. Current developments in the grouping can be distinguished into a variety of self-made/chairside hybrid Erich arch bar types and four industrially manufactured hybrid MMF systems (which we have tagged as the “League” of Commercial Hybrid MMF Systems—CHMMFSs).

This follow-up article to the Matrix Wave System (MWS) or Matrix Wave Plates (MWPs), respectively, pursues a twofold approach:To provide a compilation of the literature on hybrid MMF modalities.To analyze the design features and functionalities of commercial hybrid systems, focusing on their ability to preclude tooth root damage by the targeting function of the screw-receiving (bone anchor) holes/slots for interradicular bone anchorage; this is not about an objective comparison or conclusive ranking (‘superiority listing’), but to present a juxtaposition of the systems for self-evaluation by the readers.

## 2. Methods

This review is narrative and consulted records (studies, reports, US patents and white papers concerning hybrid MMF devices. Pertinent bibliographic references (≤January 2025) were cross-checked by keyword searching the databases PubMed and Google Scholar.

All articles underwent a full-text review. Standardized exclusion criteria in terms of quality, profoundness and evidence levels were not imposed. A notable drawback is the unusual length of the resulting descriptions and summaries.

The risk of tooth damage by screw fixation with hybrid MMF appliances from the buccal/labial side of the maxillary/mandibular alveolus may be diminished theoretically by the targeting function of the screw-receiving (bone anchor) holes/slots of the devices. In the absence of an objective metric to analyze this targeting property, the elements for bone fastening were captured in graphical schemes. These schemes outline the movability of an appliance according to all six degrees of freedom, expressed using targeting boards.

For a brief review of screw insertion sites, the abstracts of the most relevant publications were screened and a representative selection of full texts was excerpted to provide the elements of the discussion. A full print version of this review would inevitably be extensive. Consequently, several sections are provided as supplementary electronic content for optional consultation.

## 3. Results

### 3.1. Review—MMF Appliances

The core objective for hybrid MMF appliances was blending the assets of conventional arch bars and bone screw fixation while minimizing the shortcomings of each modality. The speed of application and reduced puncture injury risk for the surgeon coinciding with stabilization in fragmentation of the alveolar process were predominant motives for direct bone fixation of the arch bars.

Just prior to the introduction of hybrid MMF appliances, two similar case reports on the combination of arch bars and screws appeared in the literature (Tellioglu et al. 1998 [2], Gibbons et al. 2005 [3]). In both cases, the fixation of an Erich arch bar by circumdental wires in the anterior upper jaw was not possible owing to extensive prosthodontic crown and bridgework. The authors instead resorted to placing a couple of screws high up in the anterior maxillary vestibulum in order to secure the bar with twisted wire suspensions to the screws.

### 3.2. Self-Made Hybrid Erich Arch Bars—EAB Modifications

Hybrid modifications of conventional Erich Arch Bars (EABs) can be self-made, i.e., chairside-produced or prefabricated.

The modifications can consist of the simple addition of screw-receiving holes into the arch bar. The wire hooks can be unbent or the ‘winglets’ can be opened into short arms, to additionally facilitate the harboring of the holes, allowing direct bone support via transmucosal screw fixation (Figure 1). The modified hybrid EABs are not equipped with supplementary projecting beams or outriggers as abutments for fastening along the buccal/labial side of the alveolar ridges (Figure 1).

The position of the perforations for the screw fixation and reformation of the winglets vary in the modified EABs.

De Queiroz (2013) [4] interposed the perforations in the spaces between the unchanged winglets by use of a conical fissure cross-cut carbide drill bit (Figure 1, bottom left and frame A). These self-made bar modifications were mounted to the maxillary and mandibular vestibular surfaces close to the cervical portion of the teeth by predrilling with a 1.1 mm bur in the interradicular spaces and securing with 1.5 mm screws. The authors stressed not to fully tighten the screws but rather to tighten them enough to stabilize the arch bar but prevent excessive compression of the underlying mucosa. Commonly, four screws were applied (2 + 2, anteriorly and posteriorly), with more if necessary. Multiple holes were made in the bar, with only those at appropriate interradicular spaces receiving screws.

The hybrid design afforded the fixation of the arch bar in edentulous regions of the alveolar process. However, the authors noted the possibility of screw loss inhibiting durable fixation and arch bar fractures due to screw holes causing material weakness.

In a short technical note, Suresh et al. (2015) [5] suggested a simple amendment to the previously described technique to reduce fatigue fractures of the bar. De Queiroz (2013) [4] had already proposed an improved bar design, by increasing its vertical height. The additional solution consisted of opening up every second winglet and placing the perforation at its base, increasing the material surrounding the screw-receiving holes (Figure 1, frame B).

Rothe et al. (2018 [6], 2019 [7]) as well as Pathak et al. (2019) [8] offered a prefabricated modification of a conventional Erich arch bar. The number of winglets was reduced by omitting every second winglet. The enlarged spaces between the remaining winglets were available to machine screw-receiving holes enclosed by a reinforced safety ring to avoid material breakage (Figure 1 frame, C).

These prefabricated hybrid arch bars with machined screw-receiving holes were considered a superior option to self-made bars (Rothe et al. 2019) [7].

The nature of the EAB modification in a prospective clinical study in comparison with conventional EAB and MMF screws (Hassan et al. 2018) [9] unfortunately was not addressed, but the bibliographic list merely quotes de Queiroz (2013) [4] indicating the likely pattern for the EAB alteration. The paper issues a warning against weakening of the arch bar while drilling the perforations, eventually leading to premature material fractures.

**Figure 1 cmtr-18-00033-f001:**
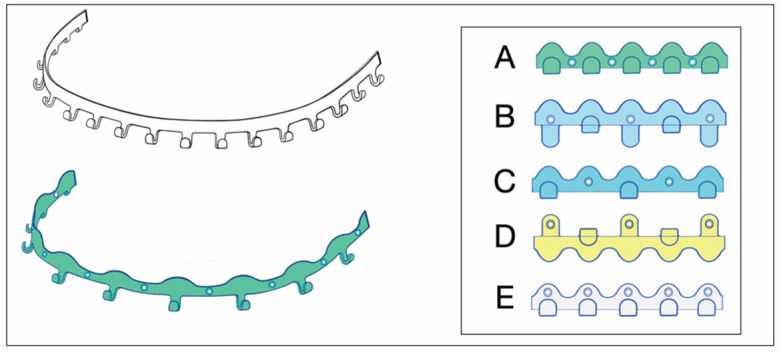
Erich arch bar modifications—(**top left**) Erich-type arch bar—for comparison. Single bar oriented for full arch application in the upper jaw—in general, arch bars come in pairs opposed in the maxilla and mandible; (**bottom left**) Erich-type arch bar modified with drill holes for screw retainment—stretched, with undulating free bar edge and reduced number of wire hooks/winglets; (**frame—right**) permutations of EAB drill perforation patterns—according to A: de Queiroz 2013 [4]; B: Suresh et al. 2015 [5]; C: Rothe et al. 2018 [6], 2019 [7]; Pathak 2019 [8]; D: Venugopalan et al. 2020 [10]; E: innominate pattern—added for completeness. (For detailed description, see text.) Source/origin: Schematic drawing—C.P. Cornelius.

A novel EAB modification variant was shown in a figure of the comparative clinical study of Venugopalan et al. (2020) [10]. Three winglets, one in the midline and two posteriorly in the premolar/molar interradicular space of a conventional EAB, were opened up like the extensions of Suresh et al. (2013) [5]. The screw-receiving openings were located, however, not at the base of the flattened winglets but in the tip of the extension arms (Figure 1, frame D). It is speculation as to whether the winglet formation was inspired by the SMARTLock Hybrid MMF^TM^ System (see later) or if it was a true reinvention. Regardless, diametrically opposed to the extensions (‘lugs’) of the SMARTLock Hybrid MMF^TM^ System, the extension arms were used to position the arch bar far out into the moveable mucosa next to the vestibular fold—predisposing it to massive mucosal overgrowth. The bar was then secured along the mucogingival line with 1.5 screws (length 6 mm) after predrilling.

To reduce compression of the underlying soft tissues and ultimately necrosis, it was suggested to screw the modified EABs gently with some free play between the metal and epithelial surfaces.

Elhadidi et al. (2023) [11] reverted to the original design modification of de Queiroz (2013) [4] (Figure 1, frame A) in their comparative controlled trial (RCT) between hybrid arch bars and conventional EABs.

### 3.3. Hybrid Arch Bars—Self-Made EAB Modifications—Clinical Studies in Comparison to Former MMF Modalities

A synopsis of the studies with clinical characteristics and outcome parameters for hybrid EAB modifications is provided as a table (Appendix A eContent) and a collection of corresponding summaries in—Appendix A.

All studies [4,5,6,7,8,9,10,11] considered hybrid/modified arch bars as an impactful innovative MMF solution, despite their differences in the comparative outcomes of EABs and MMF screws. The most frequently mentioned advantages of the modified EABs are a low risk of wire punctures, speed of application, prevention of screw loosening, application in edentulous regions and increased long-term fixation stability, all of which must be weighed against the potential disadvantages of bone-anchored MMF devices, which include irreversible tooth root damage and pulp necrosis. To reduce such injury, the use of models and appropriate imaging techniques to analyze the interradicular spaces preoperatively is recommended.

### 3.4. The League of Commercial Hybrid MMF Systems (CHMMFS)

The league of commercial bone-supported arch bars began with the launch of the Stryker SMARTLock^TM^ Hybrid MMF System (Stryker Craniomaxillofacial, Kalamazoo, MI, USA) in 2013. The Matrix Wave^TM^ Plate (DePuySynthes Craniomaxillofacial, West Chester, PA, USA) had long been in the developmental pipeline (Cornelius et al. 2025) [1] and ensued in 2014. The Microfixation OmniMax^TM^ MMF System (Zimmer-Biomet Microfixation Headquarters, Jacksonville, FL, USA) was released in 2015. Another bone-supported arch bar under the designation of the L1 MMF System (KLS Martin LP, Jacksonville, FL, USA—US Patent No. 10,470,806 B2—12 November 2019) [12] came out in North America in 2019 but has not yet arrived on the international market.

### 3.5. SMARTLock Hybrid MMF System

#### 3.5.1. Technical Features

The SMARTLock Hybrid MMF^TM^ System (Stryker Inc., Kalamazoo, MI, USA) was licensed from J.R. Marcus (US Patent No. 8118850—21 February 2012 [13], Marcus and Powers 2016 [14]).

It was the first commercially available—US FDA-approved—hybrid or bimodal MMF device on the international market with a framework made of thin malleable Titanium.

The innovative feature of a SMARTLock System is blade-like legs which project perpendicular from the hookless edge of the longitudinal arch bars. The ends of these support legs contain holes to secure the device into the alveolar bone with 2.0 optimized self-drilling locking screws of 6 mm or 8 mm length.

The supporting legs or hinges featuring a hole are regularly addressed as ‘lugs’.

The longitudinal bar segments are studded with nine supporting lugs able to serve as bone anchorage points.

The lugs are bendable and can be adjusted to face their openings over the interradicular spaces for transmucosal monocortical screw insertion.

The locking mechanism is obtained by a bone screw with a threaded conical locking head below the screw head and above the screw shaft. The screw-receiving hole in the lugs accommodates a single matching top countersunk hole into which the cones of the screw engage successively during insertion to hold the plate at an elevated level above the oral mucosa, creating a “Standoff”. A plate spacer fork can be used for assistance. Currently, the SMARTLock Hybrid MMF^TM^ System bar element comes in a regular size (gold-colored) with relatively prominent lugs (Figure 2A,B) and a smaller, low-profile version (silver-colored) with attenuated, less projecting, lugs (Nizam and Ziccardi 2014 [15], Chao and Hulsen 2015 [16], Kendrick et al. 2016 [17], Kendrick and Park 2016 [18], Stryker SMARTLock Brochure—Last internet access February 2025).

The position of the openings in the lugs allows leaning the arch bar against the tooth equators in the manner of a fence and to bridge the attached gingiva up to the mucogingival line. The construct is then secured with screws, ideally inside the zone of the attached gingiva.

#### 3.5.2. Mode of Application

As with conventional arch bars, the SMARTLock Hybrid MMF System is manually contoured and trimmed according to the maxillary or mandibular arch dimensions before mounting. The placement of screws starts in the midline of the respective jaw and progresses laterally toward the molar region. The locking mechanism, if precisely engaged for each plate hole–screw connection, maintains an appropriate standoff between the undersurface of the arch bar and the mucosa.

Acknowledged indications are the temporary (intraoperative and short-term postoperative) stabilization of mandibular and maxillary fractures to their preinjury occlusion in patients with erupted adult dentition (≥12 years old).

#### 3.5.3. SMARTLock Hybrid MMF System—Clinical Studies

A range of studies on the clinical usage of the SMARTLock hybrid MMF device exist in the literature. In the last decade, 15 specific contributions have been published. These include one pilot study (Nizam and Ziccardi 2014) [15], one early report (Kendrick et al. 2016 [17]), seven comparative studies on conventional Erich arch bars (Chao and Hulsen 2015 [16], Rani et al. 2018 [19], Bouloux 2018 [20], King and Christensen 2019 [21], Khelemsky et al. 2019 [22], Sankar et al. 2023 [23], Burman et al. 2023 [24]), two comparative studies on MMF screws (Roeder et al. 2018 [25], Aslam-Pervez et al. 2018 [26]), two comparative studies on conventional Erich arch bars and MMF screw fixation (Edmunds et al. 2019 [27], Salavadi et al. 2025 [28]), one abstract investigating the risk of dental injuries (Wilt et al. 2019) [29] and one technical note on the applicability in edentulous conditions (Carlson et al. 2017) [30].

Jain et al. (2021) [31] aggregated a meta-analysis comparing bone-supported arch bars to EAB controls in a limited selection of seven randomized trials: four chosen from the SMARTLock group (Rani et al. 2018 [19], Bouloux 2018 [20], King and Christensen 2019 [21], Khelemsky et al. 2019 [22]) and three from self-made constructions (Hassan et al. 2018 [9], Pathak et al. 2019 [8], Rothe et al. 2019 [7]).

Sulistyani et al. (2024) [32] added a systematic review comparing treatment outcomes between tooth-borne and bone-borne intermaxillary devices. Finally, 23 studies remained, 4 of which related to the SMARTLock System (Bouloux 2018 [20], Edmunds et al. 2019 [27], Hamid and Bede 2021 [33], Sankar et al. 2023 [23]) and with 1 referring to a self-made hybrid (Pathak et al. 2019 [8]). Three papers—all concerned with SMARTLock (Edmunds et al. 2019 [27], Hamid and Bede 2021 [33], Sankar et al. 2023 [23])—were published at a more recent date than in the analysis of Jain et al. (2021) [31].

The most recent metanalysis (Kalluri et al. 2024) [34] aimed to compare the outcomes between all available MMF techniques instead of 2–3 MMF modes as had been performed in the previous literature.

A synopsis of the clinical studies on the SMARTLock Hybrid MMF System is provided in form of a table (Appendix A eContent) with three subdivisions and a collection of corresponding summaries— as Appendix A.

#### 3.5.4. SMARTLock Hybrid MMF System—Economics/Cost Analyses

This topic is outlined in—Appendix A.

#### 3.5.5. SMARTLock Hybrid MMF System—Extended Range of Applications

This topic is detailed in—Appendix A.

#### 3.5.6. SMARTLock Hybrid MMF System—Comprehensive Appraisal

Looking at the present clinical studies, there is much potential for the SMARTLock Hybrid MMF System as a means of MMF in trauma of the facial skeleton.

The locking plate and screw design offers enhanced properties in handling combining the simplicity and speed of MMF screws with a robust, short and long-term fixation comparable to conventional EABs.

In addition, the operator’s safety was improved, because wiring is limited to a small number of intermaxillary cerclages, since no wires need to be passed through tooth embrasures. With this, the incidence of trauma to the gingivae and periodontium and compromised oral hygiene was reduced. Deficiencies and complications attributable to the bar retaining screws such as mucosal overgrowth of screw heads and tooth root damage were either considered a minor problem upon device removal or minimized with surgical experience (‘learning curve’) and considered generally preventable. It was repeatedly cautioned that tooth root injuries commonly clustered in the anterior mandible, where the roots of the lower incisors are densely aligned (Wilt et al. 2019 [29], Sankar et al. 2023 [23]).

It can be questioned why simple tooth sensibility testing methods were seldom used to monitor for possible pulp impairments/devitalization during the follow-up period for bone-anchored MMF devices.

The SMARTLock Hybrid System was predominantly compared to EAB, less frequently to MMF screws (Roeder et al. 2018 [25], Aslam-Pervez et al. 2020 [26]) and twice to both modalities (Edmunds et al. 2019 [27], Salavadi et al. [28])— (Appendix A eContent).

The indications for MMF using the SMARTLock System were not unanimous and varied widely between studies detailing closed reduction of condylar fractures or simple non-condylar mandibular fractures over ORIF of all mandibular fractures to ORIF of major craniofacial trauma including midface fractures involving occlusion as well as panfacial fractures. Roeder et al. (2018) [25] proposed limiting hybrid arch bars to facial fracture cases requiring postoperative MMF.

Despite this scope of potential indications, many author groups did not appreciate the SMARTLock System as indicated for the treatment of all kinds of mandibular and maxillary fracture patterns.

Closed reduction of simple and undisplaced mandible fractures was mostly recognized as the preferred application.

The incidence of complication rates in all three mentioned MMF techniques was comparably low, with the exception of device-specific damages.

Lip or buccal mucosa irritation by the connectors (cleats, hooks, tangs) and more recently the lugs of MMF devices is a long-known source of patient discomfort (Roeder et al. 2018) [25]. The initial high-rise profiled SMARTLock lugs were soon reduced by the company to address this situation (Kendrick et al. 2016 [17], Kendrick and Park 2016 [18], Marcus and Powers 2016 [14]).

The application time for the SMARTLock Hybrid MMF System over the cited studies showed a broad range, from a mean of 6.9 ± 3.1 min (King and Christensen 2019 [21]) to a maximum mean of 56.1 ± 15.4 min (Edmunds et al. 2019) [27], for the insertion of a total number of five retaining screws per arch in both studies. A tentative explanation may be the use of motorized screw driver equipment by King and Christensen (2019) [21]. The minimal and maximal mean placement times for EABs, 31.3 ± 9.3 min versus 98.7 ± 29.6 min, were also reported by the same studies.

The average time savings for the SMARTLock Hybrid System as opposed to EABs varied from 18.2 min (Rani et al. 2018) [19] to 42.6 min (Edmunds et al. 2019) [27]. The number of arch bar screws or circumdental wires was a key determinant for the placement as well as the removal time of the devices. Nevertheless, these numbers are oftentimes absent in publications. Two studies reported only the overall operative time including concomitant surgical procedures for the examined MMF patient groups—the SMARTLock Hybrid System compared to MMF screws by Roeder et al. (2018) [25] and to EABs by Khelemsky et al. (2019) [22].

The mean application time as well as the total operative (Roeder et al. 2018 [25], Edmunds et al. 2019 [27], Salavadi et al. 2025 [28]) time for MMF screws was 4 to 6 min shorter than for the SMARTLock Hybrid MMF System. The latter typically required more retaining screws than fixation with MMF screws.

Aslam-Pervez et al. (2020) [26] viewed this time difference as negligible and pleaded for a draw in their comparative study because the advantages and disadvantages between the two bone-anchored modalities were equally distributed.

Bouloux (2018) [20] took a rather skeptical attitude towards the SMARTLock System since the author had not observed a significant difference between hybrid devices and EABs in the overall length of surgery for isolated mandibular fractures. This was despite the fact that the hybrid installation times were shorter.

During the postoperative period, the rate of loosened MMF screws up to displacement and loss increased continually over time with detrimental effects. SMARTLock Hybrids, in contrast, offered stress shielding against the opening muscle pull on their retaining screws. This shielding was apparently generated by the locking mechanism and built up of multiple single standing screws into a durable “multi-legged” composite platform (Aslam-Pervez et al. 2020) [26].

The postoperative stability of SMARTLock bars in comparison to EABs was the subject of no more than three studies with almost coincident results. King and Christensen (2019) [21] found similar percentages of loose hardware at the time of removal for EABs (7.5 ± 10.6%) and hybrids (9.4 ±17.7%). In accordance, Sankar et al. (2023) [23] reported on good and comparable stabilities of upper and lower arch bars in hybrids and EABs.

A difficult-to-accurately-assess difference in postoperative stability scores was observed by Burman et al. (2023) [24]. A majority (85%) of patients (*n* = 20) with SMARTLock facsimile hybrid bars exhibited a high stability score category, whereas 76% of patients with EAB treatment exhibited a score categorized as unstable.

Another commonly observed adverse reaction occurring during follow-up was the mucosal coverage of arch bar screws (Burman et al. 2023) [24] as well as of isolated MMF screws (Aslam-Pervez et al. 2020) [26]. The position of screws within the mobile mucosa contrary to the attached gingiva is a crucial trigger for the inflammatory reaction. Moreover, the use of conventional screws for bone fixation with lack of a “standoff” between the device lugs and mucosa is liable to tissue overgrowth (Burman et al. 2023) [24].

So, the burying of screw heads with granulation tissue most frequently involves the anterior mandibular vestibulum, where the screws are placed inferiorly into the mobile mucosa with the intent to avoid root damage to the crowded lower incisors. Screws covered by mucosa may require minor incisions or excisions during screw/arch bar removal.

Mean removal times for hybrid SMARTLock and conventional EABs were not regularly documented in the studies. For the SMARTLock Hybrid, the mean removal time range was 10–10.5 min (Chao and Hulsen 2015 [16], Kendrick et al. 2016 [17], King and Christensen 2019) [21] to 30.1 min (Burmann et al. 2023) [24]. For conventional EABs, the removal times varied between 8 min (Chao and Hulsen 2015) [16] up to 17.9 ± 10.7 min (King and Christensen 2019) [21] and 19 min (Burmann et al. 2023) [24].

SMARTLock bars were less likely to need removal under general anesthesia than conventional EABs (Edmunds et al. 2019) [27].

The often retrospective nature of these studies is the reason for deficits or gaps in data of parameters, which need continuous follow-up observation and recording.

### 3.6. OmniMax^TM^ MMF System

#### 3.6.1. Technical Features

The OmniMax MMF System (Zimmer-Biomet, Jacksonville, FL, USA) is a bone-anchored MMF system and is composed of preformed arch bars (plates) and locking screws based on the same basic principle as other hybrid systems (Figure 3). The market release was in 2016.

The embodiment of the OmniMax arch bars is an in-plane-bent or curved plate, respectively, that carries 12 horizontal slots rising along its length (Figure 3) to accept the bone retaining screws (OmniMax Device ID K143336, Biomet Microfixation OmniMax MMF System, US Patent No 2015/0297272 A1—22 October 2015 [35], Zimmer OmniMax Brochure—Last Internet access February 2025). These screw slots have an elongated oval shape and are organized inside six mounting tabs, each of them enclosing a pair of such slots with a supporting strut in between. Five U-shaped notches between the mounting taps interrupt the longitudinal plate profile and provide a segmental geometry of six uniform sections.

The plate edge opposite to these sections is equipped with 12 hooks at regular intervals for the intermaxillary wire cerclages. The hooks form a J-shape with an elongated leg between the base and the turn-up ending.

A duo of hooks next to the outer pillars of a plate section recurrently contributes to the serial pattern of the embodiment.

The locking screws are manufactured from Titanium Alloy (Ti-6Al-4V) in the design of 2.0 self-drilling screws in three lengths: 7 mm, 9 mm and 11 mm. The screws have a shaft with a first set of threads for bone insertion and anchorage. A second straight threaded portion with a larger outer diameter is found between the screw head and the screw shaft for adjustable locking into the plate. An annular groove is incorporated directly below the screw head.

The pitch of this bone thread set is twice the pitch of the second set of straight threads, thereby giving the bone threads twice the lead of the second thread portion.

The core diameter of the annular groove is larger than the diameter of the second thread set and is sized for a tight fit within the beveled rims of the slot aperture of the plate (arch bar) (Figure 3 and Figure 4). These properties of the OmniMax screw are the basis for the plate’s standoff feature, which creates a gap between the plate and the gingivo-mucosal surface.

During continuous insertion of an OmniMax self-drilling screw, the second thread portion will engage the rim of the slot aperture. Owing to the different lead of the threads, with further advancement into the bone, the plate will be raised towards the head of the screw until the rim of the slot will be finally seated into the annular retention groove of the screw and held there by tight friction. For the plate, once seated within the screw’s locking groove, the standoff height can be tuned by turning the screw out of or into the bone. The gap can be further established and adjusted without a spacer tool.

The in-plane pre-bent curvature of the plates (arch bars) and the segmentation at the side of the notches conceivably facilitates the contouring to the anatomy of the mandible or maxilla.

#### 3.6.2. Mode of Application

The arch bars are manually molded to approximate the anatomical conditions and trimmed to the appropriate length. The plates are positioned close by the gum line so that the hooks will be placed vertically at the level of the tooth necks, interdental papillae, marginal epithelia and sulci, respectively. The single sections of the plates need consecutive angulation and/or rotation against each other once localized at the notches to target the slots over risk-free alveolar screw insertion points. The action and reaction principle must be considered when reorienting the sections.

Within the aperture of a slot, the screw position can be slid horizontally across a range of about 5–6 mm. Depending on its length, a minimal number of four screws per bar (Figure 3 and Figure 4A,B) or of two for a single section bar-segment (Figure 5A,B) is required to provide rotation stability.

#### 3.6.3. OmniMax^TM^ MMF System—Clinical Studies

To date, there is only one completed clinical study comparing the OmniMax MMF System with conventional Erich arch bars (EABs) (Aukerman et al. 2022) [36]. More precisely, this retrospective chart review compares 23 patients treated with the OmniMax hybrid with 18 patients having received EABs. The demographic data of both groups were homogenous. The assessment parameters were the mean total duration of surgery (surprisingly not the time for installation of the MMF devices!) and short-term complications including unexpected return to OR, 30-day postoperative infection rate, neuropathy, malocclusion and facial contour deformities. The indications for surgery were not reported. The mean total operating time was 84.9 min with OmniMax compared to 96.6 min with conventional EABs; this difference was not statistically significant. None of the short-term complications differed between the patient groups. The largest difference was exhibited for malocclusion—occurring in 9% (2/22) patients after the OmniMax MMF treatment versus 22% (4/18) patients with EAB.

A US clinical trial registration of a single-cohort study for clinical evaluation of the OmniMax MMF System (ClinicalTrials.gov ID: NCT03075865) dates to 2017. It was conceived as a multicenter prospective observational clinical trial. The investigative aim was to evaluate the efficiency of the OmniMax Hybrid MMF System in ORIF of mandibular fractures. Patient enrollment began in June 2017. A brief interim report presented outcomes for 19 patients (Morio et al. 2018) [37]. The mean application time for the OmniMax MMF hybrids was 12.8 ± 3.0 min. The average postoperative wearing period of hybrid MMF assemblies was 51.1 ± 9.7 days. Healing was uneventful in all cases.

Regular oral hygiene screening during the MMF interval showed that 78.9% (*n* = 15/19) had maintained or improved hygiene.

The time for removal of the devices was to 2.7 ± 1.2 min.

No glove perforations or accidental skin punctures during device application or removal were documented.

Postoperative CBCT analyses showed no screw contact in 91% of 300 tooth roots, whereas 8.3% had minor root contact and 0.7% had major root contact but without need for further treatment.

The authors concluded that interradicular screw insertion can be accomplished with minimal risk using appropriate preoperative imaging.

At the final visit, 15.8% (*n* = 3/19) of cases had injured periodontal structures, 1 case (5.2%) demonstrated mucosal screw (device) overgrowth and there was no gingival necrosis.

In terms of Quality of Life (QoL) metrics, the patients had minimal complaints at the end of treatment: the mean comparative pain score (0–10 scale) decreased from 5.21 preoperatively to 1.89 postoperatively prior to device removal.

Meanwhile, the recruiting phase (39 patients enrolled) for the trial was completed. The last study update was submitted to ClinicalTrials.Gov in July 2021 and the final results have not yet been posted (current status: February 2025) or published.

### 3.7. L1 MMF System (KLS Martin)

#### 3.7.1. Technical Features

A recent arrival to the league of bone-borne (hybrid) MMF systems is the L1 MMF Device (KLS Martin), consisting of single-edge toothed titanium arch bars and plural slider plates, which are affixed to the dento-alveolar bone with self-drilling locking screws. The present design (Figure 5A–C) corresponds to an optimized or alternate (2nd or V 2.0) version of the device in the United States (US Patent No. 10,470,806 B2—12 November 2019) [12] with changes to the mounting/slider plates (Figure 6). The L1 MMF Device was released to market in North America in 2019 and is currently (2025) awaiting commercial launch in Europe.

This KLS Martin L1 MMF System is made from Titanium and consists of rack-edged arch bars and relocatable slider plates with a 0.5 mm profile. The arch bars come in a “7-hole” and a “9-hole” length with seven or nine slider plates which contain the holes for bone-anchorage with a locking screw. The rack edge of an arch bar contains seven or nine segmental rows of spaced rectangular tabs and gaps which are separated by six or eight wire hooks, respectively. The edge of the central section has an even surface at both sides of a midline indicator (Figure 6). This indicator resembles a trident with two extended outer processes and an inner tab. In the upcoming international market versions, the rack edges will be varied in terms of the total number of wire hooks, their mutual proximity and design of the spaces in between (evenly or regularly jagged by tabs and gaps).

Each slider plate comprises three parts—a coupling portion extending from a main body and a screw-receiving hole (Figure 6 and Figure 7)—building up a front side, an intermediate U-shaped plication and a rear side, which abuts the mucosal tissues.

The coupling portion has a large hourglass-shaped aperture within the front and backside. The narrow neck of the hourglass corresponds to a pair of legs bordering an elongated vertical slot. This configuration provides an improved overview and even insight of the leeway space of the coupling portion. This permits better control of plate maneuverability, adaptation and engagement into the arch bar.

Each section of the arch bar has a single slider plate, which can be slid or shifted in the transverse direction along the row of tabs and gaps within the confines between two wire hooks. The interdigitation of the slot and legs of the plates’ coupling portion with the tabs and gaps (‘teeth’) of the rack edge enables some moveability, so that the plate can be twisted, tipped over and/or angulated.

#### 3.7.2. L1 MMF System—Mode of Application

The application of an L1 MMF bar begins with the selection of an appropriate bar length and shaping it to the patient’s anatomy. If there is no fracture interrupting the dental arch, the center of the bar, indicated by the trident and/or a laser etching, is positioned in the midline of the upper or lower jaw. The bar’s spatial orientation is directed alongside the mucogingival gum line and is intended not to cover the free gingival margins (L1 MMF Technique guide and catalogue, KLS Martin; You Tube Videos—L1^®^ MMF, Technique Guide; L1^®^ MMF, Surgical Technique). The slider plate is then released from its coupling in the trident. The loose slider plate is moved on all three axes to scan the alveolar relief with the screw-receiving hole until an appropriate interradicular space is found.

As soon as a safe location is identified, the leg extensions of the slider plate are seated on the tab of the bar and the anchoring screw is inserted. The slider plates are secured with 2.0 mm self-drilling locking screws of 6 or 8 mm lengths (Figure 5A–C).

The locking mechanism between the screw and slider plate is not yet activated. Prior to activation, the slider plates along the laterally neighboring bar sections are slid sideways into position, engaged into the rack profile and provisionally fastened with screws.

It must be noted that room for up-and-down mobility of the bar would persist if the slider plates were aligned perpendicular and in parallel to the bar unit, since the bar could enter and back out of the leeway spaces inside the coupling portion of the plates, alternatingly. To immobilize the arch bar section, the ‘teeth’ of the rack edges must immutably stack within the slot extension of the slider plates.

A reliable foothold between every two successive bar sections and the associated pair of slider plates can be effectively achieved by fastening the plates in a mutual angular position, pulling the bar teeth into the plate slots.

For a tight anchorage of the remainder of the bar sections towards the molar regions, each following slider plate must be coupled not in parallel but in a diverging angulation to the previous one (Figure 5A,B).

For continuous control and to enable modifications during this fastening procedure, the screws are not definitely locked into the slider plate until the setting is reliably pre-assembled.

A bar bracing the entire mandibular or maxillary arch will require five mounting/slider plates to hold it in position. At the end, the maximal rigidity of the overall MMF system is leveraged as soon as intermaxillary wire ligatures are applied to the hooks and fully tightened in dental occlusion (Figure 5A–C).

The L1 MMF System is designed to maintain proper occlusion temporarily during fixation of mandibular and maxillary fractures and during postoperative bone healing for up to 6–8 weeks.

The system includes some specialized instruments such as a plate spacer fork, a bar cutter and a ligature tucker.

#### 3.7.3. L1 MMF Device (KLS Martin)—Clinical Studies

No white papers or clinical studies on the L1 MMF device have been identified in a search of the Cochrane and Pub-Med literature databases.

### 3.8. Matrix Wave^TM^ System (DePuySynthes)

#### 3.8.1. Technical Features

The technical features of the MWS (US Patent No.: US 9,820,77 B2—23 June 2018) [38] have been outlined in great detail already (Cornelius et al. 2025) [1] and are not repeated here.

#### 3.8.2. Mode of Application

The mode of application has been described in great detail already (Cornelius et al. 2025) [1] and is not repeated here.

#### 3.8.3. MatrixWave MMF System (DePuySynthes)—Clinical Study

Up to now, just a single case study illuminates the clinical issues of Matrix Wave Plates (Kiwanuka et al. 2017) [39]. This study outlines a series of eight consecutive patients who sustained two concurrent mandibular fractures. Three patients were reported in detail, including pre- or postoperative CT and/or panoramic X-ray imaging.

Closed reduction to reestablish preinjury occlusion was performed first, followed by installation of the Matrix Wave System (MWS). In the mandible, attention was paid to place the Matrix Wave (MW) locking screws in direct proximity to each side of the fractures where possible. 6 mm screws in length were used in the mandible and 8 mm screws were used in the maxilla (length specification given according to manufacturer). The number of screws inserted per arch was not reported. As shown in an illustrative postoperative orthopantomogram, the plate holes in the maxilla were only locked with screws at the vertical bony pillars right below the piriform rims and the base of the zygomatico-maxillary buttresses, while the holes in between were unoccupied. In contrast, all mandibular plate holes were filled with screws.

The recesses (height 1 mm) below the prominent cap-shaped screw heads were used to apply wires across the fracture line with the goal of better approximation of the fragments and the introduction of compression (analogous to bridal wires).

Intermaxillary wire cerclages were mounted around single or pairs of opposing MWS hooks/tie-up-cleats, occasionally in combination with a screw head recess. The average treatment duration of intermaxillary wire fixation was 6 weeks. If indicated, guiding elastics were applied for an additional 2 weeks until hardware removal in an office setting. The time required for the application or removal of the MWPs was not reported in the article.

Intraoperative wire-stick injuries were not experienced. Postoperative imaging did not show any tooth root damage.

There were no major complications in the postoperative treatment course.

Preinjury occlusion was reestablished in all patients. No laxity or instabilities of the Matrix Wave units were recorded, and there were no clinical or radiographic signs of bony malunion or nonunion.

Oral hygiene was not compromised in any patients according to the study.

Mucosal overgrowth and embedding of the screws were also not observed.

The authors concluded the MWS offers advantages for the closed treatment of two or multiple concurrent mandible fractures because of two unique features, the horizontal malleability of the bar unit and the prominent screw head/recesses that allow for placement of intermaxillary wire cerclages.

### 3.9. Juxtaposition of the League of CHMMFSs

The individual commercial hybrid MMF devices have gained varying levels of popularity and clinical adoption over time.

The SMARTLock Hybrid MMF System from Stryker is the most widely known device, also having the longest lifespan on the market since 2013. It is also the most frequently and intensely discussed of the hybrid MMF systems.

There is a paucity of information on clinical outcomes for the other three hybrid MMF systems. A juxtaposition of the league members’ design will illustrate their technical features and functionality.

#### 3.9.1. Common and Distinguishing Technical Features—Embodiments, Design and Targeting Functionality

The conventional Erich arch bar is the origin for the design and functional construction for three hybrid variants: the SMARTLock System, the OmniMax System and the L1 MMF System.

##### SMARTLock MMF System

The flat band shaped configuration with incorporated hooks along one edge of the Erich arch bar serves as the blueprint for the embodiments of the three systems (Figure 7, Figure 8, Figure 9 and Figure 11).

This becomes overt at the first glance at the SMARTLock Hybrid bar (Figure 7).

The conventional Erich arch bar serving as the basic structural SMARTLock constituent is equipped with multiple “lugs” (Figure 2A, Figure 7 and Figure 8).

To target the desired point for insertion of a bone screw, the lugs can be bent around an arch of rotation. Moreover, the lugs can be extended along their vertical axis by either flattening or steepening the angled footing portion (buckling) of the lug. Doing so, however, can alter the standoff from the mucosal surface.

The range of these movements is controlled by the vertical height of the lug and its material properties. Geometrically, these potential movements cover a crescentic surface area (Figure 8).

Theoretically, the use of a short SMARTLock segment with a single lug could be moved in space to meet with an appropriate screw insertion site. However, the stability of this construct would need evaluation.

For larger segments with two or more lugs, the position of the bar element in space is flexible, whereas the movement of each lug underlies the delineated principles.

##### OmniMax MMF System

The outer design feature of the OmniMax MMF System is the extended horizontal screw insertion slots, embedded pairwise within a row of suprastructures, with each of these structures reminiscent of a railing on three posts.

These railings abut on the hookless border of an Erich-type arch bar, which again serves as the fundamental element for the device (Figure 9). A series of six railings tabs segmentalizes the basic bar and provides five pivot points in between.

A systematic analysis of the range of movements to allow safe screw insertion sites with the OmniMax arch bar can become quite complex (Figure 10). In fact, the horizontal slot pairs and the pivoting bar segments interact and are interdependent. The movements of a single segment with two combined screw insertion slots are still clear and simple. The segment can be moved three-dimensionally, which will affect the position of the horizontal slots. Finally, the screws are slid inside the slots to a riskless insertion point. If a single segment is used, it needs to be secured against rotation with two screws—one in each slot (Figure 9 and Figure 10).

With movements of a succession of segments (≥2 or more), the intermediate junctions act as articulations, allowing for pivoting around all three axes of rotation. Complicating the process further, any spin movement around an axis will indirectly affect the next (third) adjacent segment and carry it along the same direction in a chain reaction (Figure 10). This process again entails the need for compensating motions at the next junction. In a construct of several successive segments, a single screw in either slot of a segment is sufficient for fixation of the entire device. This means in a full arch of five or six segments, three or four segments are fastened at the posterior endings and in a median or paramedian position with a screw, while the intermediate segments remain empty.

##### L1 MMF System

The L1 MMF System is designed as a multicomponent system consisting of a fundamental bar unit derived from a conventional Erich arch bar and an ensemble of vertically arranged members, the mounting or slider plates (Figure 11). The rack edge of the bar unit is segmentalized by intermediate wire hooks into consecutive rows toothed with rectangular tabs and gaps. The central segment of the bar unit contains a trident process (Figure 5B and Figure 6) with two outer appendages and an inner rectangular tap formation. The elongated trident in the middle of the bar unit serves to define the midline of the dental arch as well as leveling the bar at an appropriate vertical height.

The rack profiles on either side of the central segment provide transverse guiding lanes for the slider plates.

The slider plates have a two-leg slot extension to snap into one of the continuous rectangular “tab and gap” (=tooth) spacings along with the rack edge in the bar sections, similar to a gear mechanism. The back and front side of a plate’s coupling portion enclose the bar unit like an oval tube (Figure 5A, Figure 6 and Figure 11).

One slider plate is pre-assembled per segment. Therefore, two chained segments are the shortest applicable partitioning of an L1 MMF bar.

**Figure 11 cmtr-18-00033-f011:**
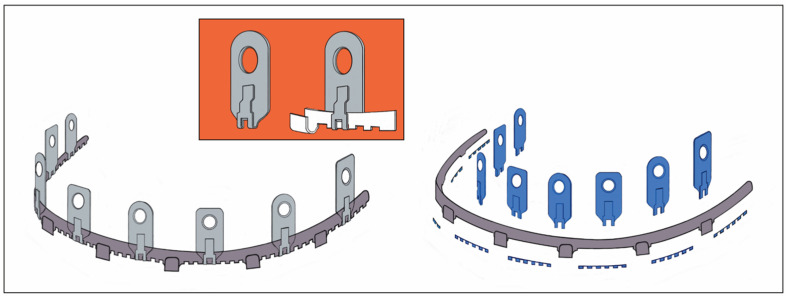
Schemes of an L1 MMF device—oriented and conformed in curvature for maxillary application—stylized schemes of the embodiment. (**left**) Assembly of arch bar unit and multiple mounting/sliding plates. Sliding plates serve as footholds for the arch bar and for bony anchorage. (inset—orange) Sliding plate (simplified initial version) with screw-receiving hole and coupling unit. (**right**) Blow-up/cutaway illustration detailing the features which convert a conventional dentally fixed Erich arch bar into the bone-anchored L1 MMF device: tabs and sliding plates. Source/origin: 1st version for the embodiment in Patent No. US 10,470,806 B2—12 November 2019 [12]. Schematic drawing—C.P. Cornelius.

The simplest way to remove a slider plate is by shortening the bar medial to a wire hook stop. The slider plate can then be removed from the bar and must not be cut and bent open.

An L1 MMF bar unit—anatomically adapted in length and curvature—is applied via the slider plates, which can move transversely. Each plate is brought into a position, in which the screw-receiving hole lies above a safe interradicular space and the two-leg slot is engaged within the rack.

The mechanical interdigitation (=plug connection) between the tooth spacing of the rack and the slider plate leg extension is not a high-precision fit but rather a contact relationship, leaving some degree of flexibility in any direction. The slider plate can be twisted, tipped over or angulated within given constraints along the three rotational axes (Figure 12).

A plural (≥2) of slotted slider plates fastened under tension in diverging vector directions is required to immobilize partitions or the entire bar unit. Otherwise, the bar unit could uncouple from the slots and shift relative to the slider plates inside their interior leeway spaces (Figure 6 and Figure 7). The slider plate slots are rigidly locked into the toothed spacings of the bar during the final wiring of the intermaxillary ligatures.

##### Matrix Wave^TM^ System

The “plates” of the Matrix Wave^TM^ System (MWS) differ appreciably from conventional Erich arch bars. The square-shaped rod profile extending in the pattern of a periodic sinus wave does not compare to the band-like framework of the Erich arch bar (Figure 13). The rod is light, quite soft and flexible, so that it is easily adapted to the dental arch.

The screw-receiving (bone anchor) holes, integral to the MWS embodiments, are located in short flattened portions (‘plateaus’) within the rod which alternate with the raised Omega curve partitions.

By contrast, in Erich-type hybrid devices, the holes for the bony connection are contained in vertical projections such as lugs, railings or stackable slotted slider plates.

The shortest clinically useful MWP subdivision includes the Omega-shaped segment with its tie-up cleat and the two screw-receiving holes (Figure 13 and Figure 14).

A one-wavelength Omega segment can be oriented, molded (stretched apart or pinched) and adjusted in every conceivable way to bring the cleat into an appropriate position next a tooth equator in the premolar/molar region of the upper or lower jaw while concurrently placing the screw-receiving holes over an interradicular space at the mucogingival gum line.

To determine the targeting functionality of a screw-receiving hole within a pluri-segmental MWP partition, an appreciation of the possible movements in—at least—two consecutive wavelength MWP sections (i.e., two adjacent Omega segments) is necessary (Figure 14).

The spatial MWS pattern can be transformed with distinct bending movements and balancing motions to bring the screw-receiving hole to a safe and controllable site or move a cleat attachment into a favorable position along the tooth equators.

#### 3.9.2. Bony Fixation/CHMMFS Retaining Locking Screws

Bony fixation of the hybrid arch bars or MMF devices at the level of the attached gingiva above or below the mucogingival gum line is essential to prevent or reduce granulation tissue and mucosal overgrowth of the screw heads.

This feature is principally workable for all four CHMMFS. The overall height profiles of the devices do, however, vary, and a low size vertical embodiment may not be suitable to bridge the distance between the gum line and the tooth equators. Here, the hooks or cleats for the intermaxillary ligatures are ideally positioned for biomechanical efficiency (Cornelius et al. 2024, Part I) [1]. An upward or downward shifting of the bony fixation level brings changes to the relative position between the bar units of the hybrid devices and the gingival margin, the tooth necks or the lateral crown surfaces. In the worst case, the flat band of the bar unit covers and possibly impinges large areas of papillary and free marginal gingiva. This can lead to impaired oral hygiene and, with food impaction, debris and plaque accumulation.

The attached gingiva has a keratinized epithelial surface, which is better suited for these mechanical stresses than the mobile alveolar mucosa.

The width of the attached gingiva on the buccal/labial side can vary greatly between individuals and within an individual relative to tooth positions within the dental arch. With healthy periodontal conditions the maximum amounts of the buccal/labial attached gingiva is to be expected along the maxillary and mandibular incisors (lateral > central), followed by the canines, first molars, second and first premolars (Ainamo et al. 1981 [41], Anand et al. 2022 [42]). The attached gingiva narrows next to the frenula and buccinator muscle attachments (Ainamo and Löe 1966) [41]. Altogether, maxillary teeth have wider zones of attached keratinized gingiva than the corresponding teeth in the mandible. There are no gender-specific width differences (Ainamo and Löe1966 [43], Jennes at al 2021 [44], Anand et al. 2022 [42]). The zone of the attached gingiva widens in adult age with the most progression between the third and fifth decade of life (Ainamo et al. 1981) [41]. Thick periodontal phenotypes seem to be associated with a greater width of the attached gingiva (Vlachodimou et al. 2021) [45].

A vulnerability of the different periodontal phenotypes (thin versus thick attached mucosa) to long-term penetration by the hybrid MMF retaining screws has not been explored in the literature.

Conventional screws to fix hybrid MMF devices risk compressing the gingival soft tissues but also trigger disproportionate mucosal overgrowth, as exemplified in several studies (Rani et al. 2018 [19], Sankar et al. 2023 [23], Burman et al. 2023 [24], Appendix A eContent).

The use of locking screws for fixation of hybrid MMF devices provides a hovering or standoff property, so that compression and necrosis of the gingival soft tissues can be inhibited. A secure standoff can also be maintained by use of a plate spacer or dedicated tool, creating a stop between the mucosa and the plate as the locking screws are tightened.

At first glance, the design of the locking screws of the different CHMMFSs does not show major differences concerning the bone threads and the conical locking heads with their secondary reverse threads, which are secured into the plate holes (Figure 15).

There are, however, several distinctive features that play key roles in the locking mechanism (e.g., insertion at variable angles), the standoff function and the overall stability of CHMMFSs—for details, see—Appendix A.

In point of fact, the vertical height of the conical locking heads differs and decreases in the following order: OmniMax^TM^ MMF System (Zimmer Biomet)—2.5 mm > SMARTLock^TM^ System (Stryker Inc., Kalamazoo, MI, USA)—1.5 mm = Matrix Wave MMF System—1.5 mm > L1 MMF System (KLS Martin)—1.4 mm.

In principle, higher locking heads enable larger spaces between the plate/bar and the mucosa. The overall design and geometry of the interface between locking heads and screw-receiving hole is certainly a cofactor in predicting standoff.

How much standoff distance is adequate to prevent adverse tissue effects is not yet determined but might be forthcoming before long. The width of coverage of the periodontal transition zone to the tooth necks by the extent of the bar/rack/device will also be implicated in this problem.

At present, it is recommended that the maximal standoff in clinical situations is obtained.

For further information on the technical features of the CHMMFS screws, see—Appendix A.

#### 3.9.3. Locking Screw Differences

The OmniMax and L1 MMF screws have small annular grooves underneath the screw heads.

A noteworthy feature of the Matrix Wave locking screws is a relatively large recessed neck underneath the prominent cap-shaped head. The MWP locking holes are engaged in the plateaus below the recess level and keep the neck areas available for bridal wires, supplementary intermaxillary cerclages and/or elastic loops—in many connection variants (see Figure 16)—plus screw-to-screw elastic loop, cleat-to-cleat wire cerclage, two screws-to-cleat elastic, loop or wire, and vice versa.

The screws of all four HCMMFSs have raised flat disk-like or cap-shaped heads with cruciform recesses, which are countersunk in the L1 MMF and the Matrix Wave locking screws.

All hybrid CHMMFS locking screws are self-drilling.

The manufacturers’ specifications in the brochures and technical guides are limited in details of the technical screw parameters such as thread angles, minor thread diameters, pitch angles, pitch diameters or lead and their impact on the lifting performance and on the locking mechanisms. For details, see —Appendix A.

The fixation of the hybrid MMF bars/devices with an increasing number of locking screws will progressively enhance the robustness of the monobloc framework and produce a hardware and bony structural unit that is able to resist pull out forces and slippage.

Failure of the monobloc can occur if the locking screws have been inaccurately or loosely engaged into the beveled or threaded holes/slots of the bar or rod, respectively.

In mechanical overload, the tightly locked screws will either break or dislodge from the bone combined with the framework.

In contrast with conventional screw fixation, the load accumulates serially and the pull out proceeds sequentially one screw at a time.

#### 3.9.4. Cleats

The design features of the cleats of the various MMF hybrid systems differ in terms of length, strength, configuration (rod versus winglet), curvature, bendability, insertion at the embodiment (bar), number and distance from one another. The particular design may offer practical advantages, especially providing greater ease when carrying out belaying with wire ligatures or traction elastics in the posterior oral cavity.

## 4. Discussion

Arch bars or MMF devices with a hybrid or bimodal character can be defined as a blend of conventional arch bar or rod-shaped embodiments outfitted with elements (e.g., plate/screw-receiving holes, lugs, slots) to provide direct bone support by means of screw fixation laterally in the alveolar process.

Self-made hybrid arch bars and, above all, the commercial versions of hybrid MMF devices are often dubbed “easy arch bars” to underscore that they address some major drawbacks of tooth-borne conventional arch bars (EABs)—these include the time requirements for application and removal, the risk of wire-stick injuries and periodontal impairment/damage in long-term applications.

Because hybrid arch bars/MMF systems are likely to gain increasing importance in facial trauma care because of potential advantages over former techniques for jaw immobilization, this review explores the current landscape. All available publications (studies, reports, US patents and white papers) on the subject have been considered without respect to a supposed level of evidence and assessing a risk of bias analysis.

Self-made hybrid arch bars (Appendix A eContent) correspond to conventional Erich arch bars (EABs) modified by simple means—drilling additional boreholes for screw insertion and occasionally unbending or opening wire hooks, winglets or the like (Figure 1). A few publications point out their efficiency and potential clinical benefits (Appendix A eContent).

Moreover, there is a collection of CHMMFSs—the Stryker SMARTLock^TM^ Hybrid MMF System (Stryker), the OmniMax^TM^ MMF System (Zimmer Biomet), the Matrix Wave^TM^ MMF System (DePuy-Synthes) and the L1 MMF System (KLS Martin North America), as well as the latest patented hybrid MMF device (US Patent No.10,470,806 B2, 12 November 2019) [12]—which has not been rolled out internationally yet. These devices present a league of their own in terms of design, clinical performance and, last but not least, cost.

The vast majority of publications deal with the SMARTLock MMF System (Appendix A eContent), reporting the outcomes of relevant clinical parameters.

Almost all authors acknowledge the SMARTLock System as a comparable alternative to conventional Erich arch bars or MMF screws, with some even endorsing the hybrid system as the better option.

The disparity of the SMARTLock system compared to the number of publications on the other league members is great—there is only one retrospective clinical study (Aukermann et al. 2022) [36] and an interim report of a clinical trial (Morio et al. 2018) [37] for the OmniMax System, one study on the Matrix Wave System (Kiwanuka et al. 2017) [39] and no clinical data on the L1 MMF System.

The existing clinical data have been outlined in exquisite detail in Appendix A and above, so they will only briefly be revisited here.

It is unfortunate, but the poor-quality data make a comparative clinical analysis between the different hybrid MMF systems impossible.

A comparative evaluation, both among the SMARTLock publications as well between the other hybrid systems, would have been difficult from the very outset, simply because the available studies focus on different treatment methods (closed reduction versus ORIF) and different fracture patterns.

### 4.1. Wire-Stick Injuries

Glove tears and skin punctures have always been a significant concern in wire-based MMF applications.

So, the reduction of inadvertent stainless steel wire injuries to hand and fingers would be a decisive advantage of hybrid arch bars and hybrid MMF systems (Nizam and Ziccardi 2014) [15].

The rates of percutaneous injuries (superficial or deep) in wire-based MMF procedures vary between 10% and 37% (Ayoub and Rowson 2003 [46], Bali et al. 2011 [47], Chao and Hulsen 2015 [16], Osodin et al. 2022 [48]). The injuries typically affect the forefingers of the working hand (Bali et al. 2011) [47]. Traditional protectives against potential blood-borne infections include wearing double gloves and/or cerclage wires with blunt tips instead of the sharp ends resulting from mechanical wire cutters (Brandtner et al. 2015) [49]. Precut cerclage wires with round or blunted ends for a greater safety margin are now in the portfolios of medical device companies.

During intermaxillary immobilization with hybrid arch bars/MMF systems, the use of wires is confined to intermaxillary cerclages, ideally with blunt tips.

It follows logically that self-made hybrid arch bars and SMARTLock Hybrid MMF systems have shown zero or relatively low incidences of glove tears and skin punctures (Appendix A eContent). In the Aukerman study (2022) [36] on the OmniMax device, this issue was not addressed. However, the interim outcome report on a clinical multicenter trial (Morio et al. 2018) [37] documented no glove perforations or accidental skin punctures.

In the only clinical study on the Matrix Wave System (Kiwanuka et al. 2017) [39], wire-stick punctures were also absent. There is as yet no published clinical data on L1 MMF devices.

### 4.2. Speed of Application

There is no doubt the technical features of the hybrid arch bars/MMF systems and the simplicity of handling speed application and result in savings of operating time. Omitting the circumdental wiring of the arch bars, however, is not the only determinant for the acceleration of the MMF procedure. Other factors including the length or segmentation of the hybrid arch bar/MMF system, difficulty in contouring, the number of screws, the identification of a suitable screw insertion site, the maneuverability and adaptation of the bone-supporting components (lugs, slots, locking hole portion, slider plates) and manual or motorized screwdriver insertion (King and Christensen 2019) [21] may all contribute to efficiency.

As a consequence, the average time to install a hybrid MMF assembly can vary greatly. For the SMARTLock System, the time for the insertion of five retaining screws per arch ranged from a low of 6.9 ± 3.1 min (King and Christensen 2019) [21] to a high of 56.1 ± 15.4 min (Edmunds et al. 2019) [27] (Appendix A eContent). The time savings for the SMARTLock Hybrid System in comparison to EABs ranged from 18.2 min (Rani et al. 2018) [19] to 42.6 min (Edmunds et al. 2019) [27].

For the OmniMax MMF Hybrid System, a mean application time of 12.8 ± 3.0 min was documented in the interim trial report (Morio et al. 2018) [37]. There are no data on the time requirements for the Matrix Wave System and the L1 MMF System.

It is noted that the application time for pure MMF screws would be shorter than for hybrid arch bars or MMF devices, since preparation, contouring, placement and monobloc building with extra screws is omitted (Roeder et al. 2018 [25], Edmunds et al. 2019 [27], Aslam-Pervez et al. 2020 [26]).

Whenever new technologies like a hybrid arch bars/MMF systems are introduced into routine use, the time required for applications is of interest to assess economic viability. The cost-effectiveness of the SMARTLock Hybrid MMF System was scrutinized at many US centers with inconsistent results (Chao and Hulsen 2015 [16], Kendrick et al. 2016 [17], King and Christensen 2019 [21], Edmunds et al. 2019 [27]). Cost reduction can only be expected when the device cost incurred for the surgery provides a sufficient return in time savings during surgery (Khelemsky et al. 2019) [22]. The main barrier to use hybrid MMF devices on a global scale is cost. The disposable hardware carries price tags between USD 500 and USD 700 per intervention, which takes them out of reach for many surgeons in poor-resource parts of the world and can also take them out of reach for cost-restrained public-funded health care systems.

### 4.3. Segmentation

The partitioning of commercial hybrid MMF systems into short segments as proposed by our group (Cornelius et al. 2024, Part I) [1] potentially offers a cost reduction, but it depends on the technical characteristics of the segment and whether it has adequate residual stability (see above—technical features). The use of short segments of hybrid MMF systems is not indicated in complex fracture patterns (multiple fractures, multifragmentation) for certain. Determining segment lengths will vary according to the location, fragmentation and morphology of facial fractures as well as the plans for intraoperative and postoperative MMF.

### 4.4. Removal

In general, the time for removal of hybrid arch bars/MMF systems is assumed to be shorter than for wire-secured EABs, though the duration was not regularly recorded in SMARTLock studies (Appendix A eContent). Painless and efficient removal in the office is preferred. The need for anesthesia (topical, local or general anesthesia) during the systems’ removal is more important than minor differences of time, as the need for general anesthesia generates additional costs in the operating room. Obtaining access to screw heads overgrown by mucosa may require small local incisions, with longer procedure times (Hamid and Bede 2021 [33], Burman et al. 2023 [24]).

In many clinical settings, the axiom of leaving MMF ligatures in place postoperatively, for immobilization of the mandible after ORIF of non-condylar mandibular fractures, often goes unchallenged.

The reinforcement of fractures in terms of tension bending following accurate anatomical realignment and reliable stable fixation appears unnecessary from a logical standpoint, above all in the absence of specific biomechanical or clinical indications such as functional treatment with guiding elastics condylar fractures (Figure 2A,B and Figure 4A,B), multifragmentation (“comminution”) of the mandible or panfacial trauma (Saman et al. 2014 [50], Oruc et al. 2016 [51]).

Typical complications associated with immobilization of the mandible involve alterations in the masticatory muscles and the temopromandibular joints, resulting in trismus, compromised oral hygiene, impaired nutritional intake, respiratory obstruction and psychological distress—all of which can be prevented by the immediate release of intermaxillary ligatures after the completion of ORIF (Ellis and Carlson 1989) [52]. When residual instability is presumed, it seems justified to retain the MMF devices as a safeguard for potential intermaxillary fixation during follow-up.

Ellis and Graham (2002) [53] were among the earliest proponents who deliberately avoided placing their patients into “postsurgical maxillomandibular fixation” after rigid osteosynthesis for mandibular fracture surgery. The authors have not documented any associated issues arising from this omission, but they disclosed no information on whether the MMF devices were immediately removed after ORIF and intermaxillary release or retained as a precautionary measure.

In a later international multicenter RCT on the combination of rigid and nonrigid fixation versus nonrigid fixation for bilateral mandibular fractures conducted under the leadership of Ellis (Rughubar et al. 2022) [54], all patients received intraoperative MMF, and “no postoperative MMF” (i.e., intermaxillary fixation) was used, but arch bar removal was standardized to take place 6 weeks (± 7 days) post-surgery.

Occasionally, the terminology for postoperative MMF appears ambiguous—for example, Diaconu et al. (2018) [55] did not count retained arch bars for the placement of guiding elastics as postoperative MMF.

Intraoperative MMF followed by ORIF plus postoperative MMF in isolated mandibular fractures with intermaxillary immobilization periods from 5–7 days up to an average of 3 weeks repeatedly showed no difference to ORIF only (Kumar et al. 2011) [56], did not decrease complications in comparison to ORIF alone (Diaconu et al. 2018) [55] or prevented malocclusion (Roccia et al. 2022) [57].

Despite this evidence to the contrary, postoperative MMF after ORIF of mandibular trauma continues to be widely employed, as it is still believed to enhance clinical outcomes (Shenoy et al. 2011 [58], Diaconu et al. 2018 [55]).

The rationale for maintaining postoperative MMF for a few-day duration is commonly justified by various reasons, e.g., providing rest for reattachment and healing the soft tissues, TMJ immobilization and importantly to secure, adjust and settle the occlusal relations. In midface fractures involving the dental occlusion, the intraoperative intermaxillary fixation with ligatures is usually released at the completion of the ORIF as the transmission of tensile forces would have detrimental effects on the osteosynthetic fixation of the facial skeleton.

### 4.5. Screw Insertion Sites

The primary stability of a bone screw depends on adequate thickness (1–2 mm) and density of the osseous cortex, the angulation and intrabony length of the screws and the depth of insertion (i.e., monocortical or bicortical) (Brettin et al. 2009 [59], Yang et al. 2015 [60]). The search for the most suitable interradicular screw insertion was fostered by the popularity of miniscrews for anchorage in modern orthodontic therapy. For details, see—Appendix A.

The preferred target sites for screw insertion from the buccal/labial side in the adult dentition—as per relevance for the placement of specialized MMF or hybrid MMF locking screws—occur in the following interradicular alveolar bone locations (Table 1).

The distance from the alveolar crest or the cemento-enamel junction towards the root tips is important to assure sufficient width and accessibility to interradicular bone spaces and the stability of miniscrews (Haddad and Saadeh 2019) [61].

Interradicular spaces in both dentoalveolar arches expand in the apical direction with the exception of the maxillary intermolar spaces, which run parallel (Pan et al. 2013) [62].

The interradicular spaces between the mandibular incisors as well as to the canines are narrow and not suitable for miniscrew insertion, so that subapical placement in the anterior lower vestibulum is recommended (Pan et al. 2013 [62], Lee et al. 2013 [63]).

### 4.6. Oral Mucosa

The type of oral mucosa along the vertical extent of the buccal/labial alveolar ridges plays a definitive role for miniscrew mid/long term failure rates.

Placement into the mobile alveolar and vestibular mucosa beyond the mucogingival line was associated with increased infection rates and failure compared to screw anchorage within the attached (keratinized) gingiva and at the mucogingival junction (Cheng et al. 2004 [64], Palone et al. 2022 [65], Xin et al. 2022 [66]).

As outlined above, screw fixation of hybrid arch bars or hybrid MMF systems within the attached gingiva up to the level along the mucogingival gum line is important in preventing mucosal overgrowth of the screw heads.

The greatest bone width of the interradicular spaces, however, does not coincide with coverage by the attached mucosa in the coronal third of the tooth roots.

The retromolar region within the confines of the maxillary tuberosity provides only a small amount of poor-quality bone.

Contrarily, the edentulous alveolar area distal to any last standing tooth in the lateral mandibular tooth row including the retromolar region and the anterior ramus offers cortical bone that is thick and very dense (Poggio et al. 2006 [67], Mehta et al. 2022 [68]).

In facial fractures affecting the occlusion, it is essential to consider the course of the fracture lines through the alveolar bone and adjacent to tooth roots. The need for direct reduction and integration of the fragments into the hybrid MMF monobloc is prioritized, resulting in the potential need to insert the screws into less favorable sites with greater risks of tooth root injury.

### 4.7. Recommendations—Screw Length and Diameter

A 6–8 mm screw length has been proposed (Deguchi et al. 2006 [69], Park and Cho 2009 [70]), which was not unopposed in favor of longer screws (Palone et al. 2022) [65]. The recommended diameter for orthodontic miniscrews for placement in the frontal/lateral alveolar bone is 1.2–1.6 mm.

Beside the miniscrew diameter, safe placement of an interradicular miniscrew requires consideration of the “peri-implant insertion path” (Ikenaka et al. 2022) [71]. This path consists of the bone/tissues between the screw and the lamina dura.

Falci et al. (2015) [72] address this insertion path as “clearance around the screw needed to preserve periodontal health and screw stability”. The authors assumed a sleeve or clearance of 1 mm adjacent to the outer (thread) screw diameter to define the measure of the insertion pathway cylinder apart from its length.

The proximity up to actual contact and its extent of miniscrews to the tooth roots is a major risk factor for long-term failure of skeletal anchorage and screw loosening. It stands to reason that if the body of miniscrews overlaid the lamina dura in their full length at postoperative control in dental X-rays, the screws had the lowest success rates compared with screws that were absolutely separate or just touching the lamina in one point with their tip (Kuroda et al. 2007) [73].

### 4.8. Imaging

The anatomic variability of the alveolar anchorage sites for screws can be great in an individual patient. When low-dose multiplanar imaging techniques reaching a sub-millimeter scale are available, the general recommendations for (mini)screw placement should not be simply relied upon (Martinelli et al. 2010 [74], Bhalla et al. 2013 [75]).

A systematic review validated distinct advantages of CBCT for interradicular (mini-)screw preinsertion planning over two-dimensional/panoramic radiographs.

The latter have limitations in estimating the size of the spaces, available bone quantity and the appropriate spatial screw orientation (Caetano et al. 2022 [76], An et al. 2019 [77]).

### 4.9. Tooth Root Injuries

Tooth root injuries are a recognized screw-associated problem and a specific complication of transalveolar anchorage of hybrid arch bars/MMF devices (Wilt et al. 2019) [29].

Radiographically, minor dental root lesions are located peripherally over the cementum and/or dentine and do not involve the pulp chamber. Major lesions project centrally onto the pulp chamber indicating a perforation, onto the entry zone of the neurovascular bundle within the root tip region or as root fragmentation.

The incidence of clinically relevant damages after the placement of hybrid arch bars (Appendix A eContent) and MMF systems (Appendix A
*eContent*) appears to be low, but it is important to recognize that minor and major dental root lesions are usually not distinguished in the reports. Several clinical studies on the SMARTLock System have not detected tooth damage in postoperative imaging (Chao and Hulsen 2015 [16], King and Christensen 2019 [21], Edmunds et al. 2019 [27], Hamid and Bede 2021 [33], Burman et al. 2023 [24], Elhadidi et al. 2023 [11]). A few other studies did not refer to tooth root injuries at all (Hassan et al. 2018 [9], Khelemsky et al. 2019 [22], Roeder et al. 2019 [25], Kiwanuka et al. 2017 [39], Aukerman et al. 2022 [36]).

The dental pulp status after dental root traumatization cannot be assessed by one single method—pulp vitality testing refers to the vascularization and pulp sensibility to sensory innervation. Radiographs confirm dental root trauma morphologically but are not diagnostic of pulp vitality or death. Commonly, sensitivity tests are employed in the form of thermal tests or electrical testing, with both methods eliciting a pain response as long as the pulp is vital and not irreversibly damaged.

Non-invasive objective testing of pulpal blood flow can be performed by Laser Doppler flowmetry, Spectrophotometry or Pulse Oxymetry, all of which requiring specialized technical equipment, training and expertise.

A reliable long-term prognosis of viability for asymptomatic teeth with radiological root damage unreactive to sensory testing is not available. Therefore, a watchful waiting approach for an injured tooth to become symptomatic is justifiable given that spontaneous repair is possible.

Three experimental studies in mature beagle dogs demonstrated an almost complete regeneration of injured periodontal structures (cementum, periodontal ligaments and bone socket) after intentional damage by miniscrew insertion (Asscherickx et al. 2004 [78], Brisceno et al. 2009 [79], Dao et al. 2009 [80]). Histologic sections of the damage sites were evaluated from 6 to 18 weeks either subsequent to the removal of the miniscrews (Asscherickx et al. 2004 [78], Brisceno et al. 2009 [79]) or leaving them in situ (Dao et al. 2009) [80]. Damage ranged from minor interruptions of cementum to pulp invasion and root fragmentation.

Restorative formation of new bone, periodontal ligaments and cementum was observed in high percentages of teeth, with increasing percentages of cementum repair of resorption zones over time. With the screws left in situ, there was no external resorption or inflammatory infiltration and/or necrosis of the pulp.

After early screw removal, abnormal types of healing were present in a third of teeth, including defective regeneration of the periodontal ligaments and bone, ankylosis at fragmentation sites and absence of repair when the pulp was affected.

### 4.10. Targeting Function—Juxtaposition of the Commercial Hybrid MMF League Members

The screw-receiving holes have received little attention as targeting facilities for pinpointing safe interradicular space for the insertion of the fixation screw.

A targeting function is very limited or nonexistent in self-made hybrid arch bars, where the row arrangement of screw-receiving holes along the rigid bar does not permit point by point alterations, only allowing for shifts or rotational movements in one piece.

The design of three of the four commercial hybrid MMF devices—the SMARTLock Hybrid MMF^TM^ device (Stryker Inc., Kalamazoo, MI, USA), OmniMax^TM^ MMF System (Zimmer Biomet) and L1 MMF System (KLS Martin)—derives from the band-like conventional Erich arch bar (Figure 7, Figure 9 and Figure 11).

The fundamental bar units of these three MMF systems are equipped with distinguishing extensions or suprastructures, to gain suspension from the maxillae or foothold in the mandible. The extensions are set up as a series of pillars or lugs (Figure 7 and Figure 8), paired slots (Figure 10) or stackable mounting/slider plates (Figure 11 and Figure 12). The suprastructures contain the screw-receiving holes or slots for bone anchorage. In contrast, the screw-receiving holes in the Matrix Wave Plates are not carried in extension arms but rather occur in a repetitive pattern in the slimline rod embodiment at regular distances along the wave length (Figure 13 and Figure 14).

The circular holes of the extensions can be moved to spot the optimal interradicular position for the screw insertion.

The heterogeneous geometry of the suprastructures among the hybrid MMF systems members, however, may restrict potential three-dimensional spatial orientation and reach. It is not possible to objectively validate and quantify these technical differences. Detailed information on the overall geometric layout, contours, measurements and material properties (e.g., malleability, elastic modulus) of the devices would be required to conduct any simulation or comparative analysis. In the absence of such data and necessary high-tech laboratory equipment or software solutions, we present graphical displays of the four commercial MMF systems. We estimate the mechanically plausible degrees of translational and rotational movements as a preliminary evaluation. These movements were depicted in crosshair target boards as a synoptic template for juxtaposition of the systems (Figure 8, Figure 10, Figure 12 and Figure 14).

This illustration was not intended to identify factual superiority of any the hybrid MMF league members or deliver a performance ranking but rather to provide a visual reference for selecting a hybrid MMF device type for routine use depending on personal preferences. The clinical utilization of the *SMARTLock MMF System* will remind surgeons of long-established routines with EABs. This familiarity with EABs is supposedly one of the reasons for the high acceptance and great popularity of the SMARTLock MMF System.

Nonetheless, the system requires the reconsideration of the vertical placement relative to the gum line. From a biomechanical perspective, the bar units and connecting hooks should be along the tooth equators. For periodontal and oral hygiene aspects, the transition zone between the tooth necks and gingival margins should be left widely uncovered. These transition zones should be bridged only intermittently by lugs or another kind of extension piece passing towards the attached gingiva.

Correspondingly, the up- or downward alignment of the SMARTLock bar/plate must be coordinated with the height of the lug extensions. Since the height is fixed, the bending radius and flexibility of the lugs is limited (Figure 8), so that compromises of the bar montage level may be needed to position screw-receiving holes over safe interradicular spaces (Kiwanuka et al. 2017) [39].

This problem becomes particularly apparent in the anterior vestibulum of the mandible, where the fixation screws are often placed into the mobile mucosa inferiorly to the root tips of the lower incisors (Figure 2A). The use of the larger (regular) of the two SMARTLock Hybrid plate versions (Kendrick et al. 2016 [17], Kendrick and Park 2016 [18], Marcus and Powers 2016 [14], Stryker SMARTLock Brochure—Last Internet access February 2025) may help to alleviate this issue.

As its name implies, the *OmniMax MMF System* possesses nearly omnidirectional bending capabilities with the prerequisite that several bar segments are utilized in a row (Figure 9 and Figure 10). A single bar segment carrying two screw slots is bendable out of plane and can be torqued. Such a segment can be displaced in all spatial dimensions to conform with the anatomy of the dental arch. The paired slot openings directly abutting alongside the bar segments (mounting tabs) provide gliding tracks for side shifting of the locking screws.

As a matter of fact, the low constructional height of the OmniMax embodiment does not allow for the placement of segments at the level of the tooth equators from the very outset (Figure 4).

Instead, the narrow two-tier design demands positioning at or over the tooth neck/gingival margin transition zone. This does not differ in concept from conventional EABs. The slots assume a targeting function only in tandem with movements of the bar segments, which set the initial framework, still with a changeable position. Then, the slots will primarily demarcate a horizontal baseline (“*x*-axis”) for potentially safe screw insertions. The screw is then slid along the slot opening to determine the final screw insertion point.

The *L1 MMF System* is characterized by a bar unit separate from a plural of slider plates (Figure 11 and Figure 12). Again, the bar unit is the fundamental element for the vertical placement of the whole assembly and as a consequence the reach of the slider plates in their passage to the attached gingiva or the upper third of the interradicular spaces. The slider plates are horizontally adjustable (Figure 5A–C and Figure 12), guided by the rack profile of the bar unit (“*x*-axis”). Once the plate foots are coupled or plugged into the rail track of the bar, the maneuverability of the plates becomes restricted to rotational movements in all three axes (Figure 12). “Sideways” movement, i.e., rotation of the plate around the *z*-axis or roll, has the greatest range; this gets less for hinges around the *x*-axis or pitch and least for twisting around the *y*-axis or yaw.

The height of the slider plate is the crucial feature for favorable vertical placement of the devices at the level of the tooth equators and along the attached gingiva in the upper third of appropriate interradicular spaces.

A particular challenge with L1 MMF installation is the fact that one slider plate alone is not enough to maintain the bar units accurately aligned and immovably fixed in place. Rather, it requires at least two slider plates in two adjacent bar sections to tighten the shortest possible bar segment. Some risk for displacement of the bar is hard to rule out, eventually.

The design of the *Matrix Wave Plate* (MWP) fundamentally differs from the other hybrid MMF systems. The straight band-like arch bar has been replaced by a continuous sinus wave design of a slender rod with a square cross section (Figure 13). The sinus wave is composed of consecutive Omega segments, which are highly moldable. The screw-receiving holes are an integral component of the rod and accommodated within small-sized tracts, the locking hole plateaus, which repeat themselves on one side of the rod at adjacent points of the same phase corresponding to a wavelength (Figure 13 and Figure 14). Each Omega segment allows for stretch, compression and torque movements and the alteration of its amplitude and symmetry.

Furthermore, plural Omega segments in a chain are pliable between one another, so that an overall more or less irregular meandering 3D configuration can be used, which comfortably reaches from the tooth equators for attachment of the connector cleats over suitable interradicular spaces. The two height versions of the MWS additionally broaden this outreach.

The individual screw-receiving holes consequently have six degrees of freedom to facilitate safe and controllable screw insertion (Figure 14).

The specific serpentine design and material characteristics of the titanium framework make it possible to realign and anatomically reduce fragments after an Omega segment has been secured with a screw on each side of the fracture line (“in-situ-bending”).

It needs to be recognized that the ideal placement of hybrid arch bars/MMF systems according to optimal biomechanical and periodontal criteria is often very demanding and at times is even not possible. Compared to individual MMF screws, applying hybrid devices is less elegant and more time-consuming (Roeder et al. 2018) [25].

An easy and comparatively cheap method to optimize the targeting function of commercial hybrid foothold extensions (i.e., lugs, slots, slider plates, locking hole sections) could be prebending them on translucent Stereolithographic models with color marking of the dental roots.

### 4.11. Tension Banding

MMF screws are frequently criticized because they do not exert a tension band function (Roccia et al. 2005 [81], Rai et al. 2011 [82], West et al. 2014 [83], Edmunds et al. 2019 [27]).

This capability is seen with arch bars, hybrid arch bars and hybrid MMF devices.

Originally, the term “Tension band” was used in connection with compression plate osteosynthesis of the mandibular body and angle fractures. Compression plating with bicortical screw fixation along the inferior mandibular border generates tension forces along the alveolar ridge or retromolar region. So, tension banding refers to a miniplate crossing, a fracture line along the superior border at the tooth root level or in the angle region to counteract splaying when compression was applied. Acrylic wire splints (Spiessl 1989) [84] or Erich arch bars (Yaremchuk and Manson 1992) [85] have also been utilized as tension band splints. In association with tooth wired arch bars, tension banding refers to the direct bracing of the teeth in the dental arch. With the advent of bone-fixed hybrid arch bars/MMF systems, this exact concept is no longer applicable. In this new setting, tension banding refers to the stress-shielding effects provided by the formation of a bloc construct or external fixator (Edmunds et al. 2019 [27], Sankar et al. 2023 [23]), which is based on the locking mechanism between the framework and bone screws.

The serpentine design of a Matrix Wave Plate is not consistent with a tension band function as a bar connection in line with the fixation screws (Figure 13). This “tension band” would be antithetical due to its flexibility and in situ bending properties. However, a transverse reinforcement between neighboring screws can be created by bridal wires or interarch cerclages in a square or “mattress” arrangement encompassing four connecting cleats (Figure 16). The stability of bridal wires or interarch cerclages compared with a flat solid bar is certainly debatable. Adjustable bridal wires can be advantageous in the closed treatment of anterior mandible fractures with the Matrix Wave System, because they allow the application of compression and manipulation while enhancing the stability without requiring the removal of the intermaxillary fixation (Kiwanuka et al. 2017) [39]. The increased stability of this “tension banding” in its original sense enables a closed treatment in cases of two concurrent mandible fractures (Kiwanuka et al. 2017) [39].

### 4.12. Transoral ORIF—Vestibular Incision Placement

All hybrid arch bars/MMF systems are compatible with transoral open reduction and plating osteosynthesis. The placement of mandibular and maxillary vestibular incisions warrants careful consideration to avoid interferences with the transmucous labial/buccal entryways of the screws and the hybrid MMF hardware. To move away from interradicular screw entrance sites, the vestibular incision should include a large cuff of mobile mucosa in continuity with the attached mucogingival tissues for effective closure (Nizam and Ziccardi 2014 [15], Roeder et al. 2018 [25]). Screw insertions into the alveolus close to the mental foramen should be controlled by bringing the incision lines curvilinear above the foramen to visualize the mental nerve branches and prevent injuries.

### 4.13. Other Associated Risk Factors and Complications

Another device-specific risk factor of hybrid arch bars/MMF systems is ischemia and pressure necrosis of the gingival and/or mucosal soft tissues caused by bone anchorage (Jain et al. 2021) [31]. The damage is preventable by using an adequate standoff between the hybrid arch bar or hybrid MMF device and the soft tissue surface. All hybrid MMF league members have a locking screw–bar/rod interface to realize an adjustable standoff. Accordingly, Morio et al. (2018) [37] documented no mucosal necrosis in the clinical outcomes of the OmniMax System. With the use of conventional screws for bone fixation, detrimental effects are most likely due to the absence of a conical locking head, leading to excessive contact pressure underneath the bar strips or device extensions. The five studies reporting on self-made hybrid arch bars (Appendix A eContent) all used conventional screws, but none even mentioned mucosal necrosis. The same is true for the three studies employing conventional screws for the bony fixation of stainless steel facsimiles of the SMARTLock System (Rani et al. 2018 [19], Sankar et al. 2023 [23], Burman et al. 2023 [24]), so that concerns regarding tissue necrosis are not yet substantiated by clinical experience.

Nonetheless, an inverse correlation between standoff and mucosa migration over the retaining hardware extensions and screw heads is supported by several findings: the rate for mucosal overgrowth within the commercial hybrid MMF league ranged from 0% (Matrix Wave—Kiwanuka et al. 2017 [39]) over 5.2% (OmniMax—Morio et al. 2018 [37]) to up to 36% (SMARTLock—Aslam-Pervez 2020 [26]) and 38% (SMARTLock—Kendrick et al. 2016 [17]).

On the other hand, with stainless steel SMARTLock facsimiles, fastened with conventional screws, rates of mucosal overgrowth ranged between 35% (Rani et al. 2018) [19], 54% (Sankar et al. 2023) [23] and 75% (Burman et al. 2023) [24].

The design of the Matrix Wave locking screws with the upper parts of the conical locking head, recess and cap-head sitting above the locking hole plateau of the rod is undoubtedly an important factor in inhibiting mucosal overgrowth (Kiwanuka et al. 2017) [39] in addition to the standoff.

Mucosal covering has no long-term adverse impact on oral hygiene. After screw and hybrid hardware removal, gingival health returns to the premorbid condition (Hassan et al. 2018 [9], Hamid and Bede 2021 [33], Sankar et al. 2023 [23]).

While hybrid arch bars/MMF devices are in use, they allow improved oral hygiene when compared to tooth wired arch bars (King and Christensen 2019) [21].

A rare, easily preventable complication, the fracture of screw heads or screw shafts, results from the application of excessive torque loads, if the screw tip hits on a tooth root and it becomes impossible to penetrate and/or bypass dentine.

Thus far, such screw fractures have not been described in combination with hybrid MMF devices. Several case reports dating from the early 2000s (Holmes and Hutchinson 2000 [86], Farr and Whear 2002 [87], Coburn et al. 2002 [88]), however, focus attention on intraoperative technical problems during the insertion of specialized MMF screws such as bent or fractured screws and shearing of the screw heads. A rare event is a fracture of tooth roots by MMF screws engaging so heavily into the radicular substance that their shafts break. Osteotomies may be required for removal of the foreign body and the tooth.

### 4.14. Screw Loosening/Postoperative Stability

An unwanted postoperative event is screw loosening.

Increasing screw failure rates, up to 24% until 5 to 6 weeks postoperatively, were reported for MMF screws used in closed or open treatment of uncomplicated mandibular fractures. The vast majority of failures occurred during closed treatment. After ORIF, the intermaxillary cerclages were released, relieving the screws from the pulling muscle forces (West et al. 2014) [83]. Many studies on hybrid arch bars/MMF systems do not report on the rate of loosened fixation screws (Appendix A, eContent). As far as the SMARTLock System, the percentages are low for locking screws, ranging between 0–0.8% (Chao and Hulsen 2015 [16], Nizam and Ziccardi 2014 [15]) and up to 12.5% (Hamid and Bede 2021) [33]. An exceptionally high number (30%) of failures (Burman et al. 2023) [24] is explainable by the use of non-locking screws. However, this rate does not appear fully consistent with the high stability scores reported for 85% of the same patient cohort.

The immediate and long-term overall stability of hybrid arch bar/MMF system assemblies clearly depends on the locking mechanism with multiple fixation screws and a bloc formation.

Very few studies, all confined to the SMARTLock System, address stability as a parameter among the clinical outcomes (King and Christensen 2019 [21], Sankar et al. 2023 [23], Burman et 2023 [24]). These studies attested comparable degrees of stability with the SMARTLock System or its facsimiles with Erich arch bars.

Periotest measurements (Schulte and Lukas 1992 [89], 1993 [90]), as used to evaluate the osseointegration of dental implants, could serve to provide objective in vitro and in vivo parameters of the stability of hybrid MMF device fixation screws in a laboratory model series or during clinical follow-up (Watanabe et al. 2017) [91].

### 4.15. Health-Related Quality of Life

The provocative title “Intermaxillary fixation—Torture or Therapy ?”, an anecdotal publication from the1980s (Mardirossian 1982) [92], gives unambiguous hints that “the subjective health status” or “Health related Quality of Life” (HrQoL) issues with MMF and any respective appliances need to be considered. Disappointingly, the publication does not discuss any HrQoL aspects but rather shares guidelines for decreasing postoperative complications during intermaxillary fixation with arch bars, suspension wires or external pin fixation. They stress a high level of oral hygiene, a high caloric diet and the removal of interarch cerclages in emergencies.

MMF has complex effects on physical, functional, psychosocial and aesthetic aspects (Nayak et al. 2024) [93]. It seems only natural that patient acceptance of MMF over a period of several weeks is low. Surprisingly, an RCT on the treatment of mandibular fractures in the tooth-bearing area using closed reduction with Erich arch bars for MMF in comparison to ORIF (Omeje et al. 2014) [94] found no difference in overall HrQoL. The treatment modalities differed in the affected domains, though: patients treated with ORIF had higher indices in the pain domain, while patients undergoing immobilization with arch bars reported higher impairment in the psychosocial and physical domains. MMF screws applied for a 6 week conservative (closed) treatment of condylar fractures during a trial (van den Bergh et al. 2015) [95] resulted in higher HrQoL scores than arch bars. The patients treated with MMF screws experienced less social isolation and had fewer problems with eating and the intake of a normal diet.

A retrospective study compared patient-reported outcomes (PRO) after the treatment of mandibular fractures in an arch bar group to an MMF screw (four- or eight-point fixation) group (Kim et al. 2022) [96]. The MMF screws provided more comfort and more favorable conditions for oral hygiene. The pain scores during MMF removal were lower in MMF screw patients.

Since the design of hybrid arch bars/MMF systems blends arch bars and bone fixation screws, the potential implications on HrQoL issues are not entirely obvious from the outset.

Currently, there exist only two systematic HrQoL investigations in hybrid MMF appliances: Pathak et al. (2019) [8] found a better QoL in patients treated with self-made hybrid arch bars than with conventional EABs, and a recent RCT (Salavadi et al. 2025) [28] pointed out superior QoL scores for MMF screws compared to SMARTLock and conventional EABs. Morio et al. (2018) [37] merely made a brief remark on decreasing pain scores upon treatment with the OmniMax MMF System.

### 4.16. Indications

Seemingly, there is no consensus about which fracture patterns optimally lend themselves to treatment with hybrid arch bars/MMF systems. Self-made hybrid arch bars, for the most part, were used for closed treatment of mandible fractures (Appendix A eContent) and in one study of maxillary fractures (Pathak et al. 2019) [8]. A single study combined self-made hybrid arch bars with open reduction and Champy–miniplate fixation for the treatment of mandible as well as Le Fort fractures (Venugopalan et al. 2020) [10].

Closed reduction of mandible fractures prevailed among the indications in the studies on the SMARTLock System as well (Appendix A eContent). Combinations with ORIF were standard in a single report (Khelemsky et al. 2019) [22] and an option in several other studies, without the reasons for selection being specified. The indication in the OmniMax trial (Morio et al. 2018) [37] as well as in the clinical series with the Matrix Wave MMF System (Kiwanuka et al. 2017) [39] was the closed reduction of single or more mandibular fractures. Maxillary/midface (Le Fort-type) fractures treated either closed (Bouloux 2028 [20], Rani et al. 2018 [19]) or open (Kendrick et al. 2016) [17] were regarded as an indication for the SMARTLock System by a minority of studies.

In general, hybrid arch bars/MMF systems are thought to be a fast, effective and robust tool for intraoperative and postoperative longer-term MMF in trauma of the facial skeleton (Edmunds et al. 2019) [27].

So, it is not quite understandable that conjunctions of hybrid MMF hardware with ORIF are underrepresented in the literature since immediate functional restoration is a central precept in the treatment of facial fractures where mandibular immobilization is recognized as disadvantageous (Ellis and Carlson 1989) [52].

We are convinced, based on clinical experience, that a suitable hybrid MMF appliance can be utilized almost invariably in ORIF of all types of facial fractures involving dental occlusion. The capability for guiding elastics postoperatively in ORIF of mandibular condylar process fractures with minimal pieces of a hybrid MMF hardware in place is just one beneficial indication.

As previously outlined (Discussion 4.6), the conditions for immediate removal of MMF devices following ORIF of fractures without mandibular condyle involvement must be assessed and, if indicated, the detachment of the devices can be completed at the conclusion of the same operative session.

Experimental model studies such as the influence of different MMF techniques on the control of occlusion in three-piece Le Fort I osteotomies (Han et al. 2024) [97] might foster similar investigations on CMF trauma and help to overcome skepticism and hesitation for some MMF methods.

### 4.17. Synopsis—The League of Commercial MMF Systems (CHMMFSs)

In the Western world, conventional Erich arch bars are still regarded as the universal standard for jaw immobilization and usually serve as the reference for MMF newcomers.

The SMARTLock Hybrid MMF System or “EAB with lugs” was developed as a spin-off and inaugurated an entirely new league of commercial MMF devices in 2013.

In the meantime, the Matrix Wave MMF System had matured into the present snake-inspired design (Cornelius et al. 2024, Part I) [1] and was launched in 2014. The novel undulating rod design with integrated screw-receiving holes afforded the system six degrees of freedom and enabled in situ bending and compression of the fragments across a fracture line. The OmniMax MMF System, introduced on the market in 2015, returned to the EAB concept and attached parallel-running slots to the flat bar for the bone-retaining screws. The L1 MMF System is the newest member, appearing on the US market in 2018, and is still awaiting global market introduction. The EAB bar unit is the basis of the L1 MMF System, which is elaborately engineered with continuous rectangular “tab and gap” spacings to plug in separate slider plates for screw montage to the alveolar bone.

The Matrix Wave MMF System deserves a special place among the commercial hybrid MMF devices, with an emphasis on its disruptive architecture and expanded functional scope, although this has not been reflected in a greater acceptance of the device. A conversion to the Matrix Wave MMF System was suggested in order to overcome the instability of isolated MMF screws (Ali and Graham 2020) [98] by the “monobloc” formation. Furthermore, the successful use of the Matrix Wave System is exemplified in two reviews of virtual surgical planning—CAD/CAM concept for repair of panfacial trauma (Sharaf et al. 2021 [99], Salinas et al. 2023 [100]). In addition, a focused registry collecting clinical data on the Matrix Wave System is underway (Liokatis et al. 2025) [101].

The simplicity and ease of application of the four commercial systems certainly varies, and none of them is immune from errors and complications. One group that was initially enthusiastic about the OmniMax MMF Hybrid System now reports that the Matrix Wave System has been implemented in their institution (Aukerman et al. 2022) [36]. The authors promoted the idea to include both hybrid systems into a future comparative study.

### 4.18. Limitations of This Review

The disparity of available data and information on hybrid MMF devices favoring the SMARTLock System is obvious. Thus, the main limitation of this review lies particularly in the extreme paucity or complete lack of clinical data on the OmniMax, Matrix Wave and L1 MMF systems. A meta-analysis in keeping with Cochrane criteria was not expected to deliver expedient results. Paradoxically, a stringent statistical analysis and anti-biased approach do not necessarily rule out misleading generalizations and inattention to important practical details (cautionary example—Kalluri 2024 [34]).

This review abstains from using exclusion criteria and instead is a detailed compilation of all admissible publications on hybrid MMF arch bars/systems in a narrative style. This has the serious disadvantage of an excessively long discussion that is not in tune with contemporary reading habits. Our review does not claim to identify a hybrid MMF variant that is superior to the others. The goal rather is to motivate treating physicians to become more familiar with modern MMF methods, including hybrid MMF appliances. In doing so, they will be able to select a system in accordance with their personal preferences, skills and the spectrum of facial trauma treated.

The evaluation of the targeting function of the commercial hybrid MMF devices is offered as a substitute for the lack of clinical data and can have a downstream impact on the prevention of tooth root injuries.

The evaluation of the targeting performance is arbitrary, however, and rests on a visual assessment of the geometric design of the devices.

A final criticism is a recognition of the authors’ bias toward the Matrix Wave System given their involvement in its developmental process (Cornelius et al. 2024, Part I) [1].

## 5. Relevance and Conclusions

Establishing mandibulo-maxillary or intermaxillary fixation (MMF or IMF) remains a fundamental component of maxillofacial trauma treatment. Despite a large number of MMF devices, there is no one single MMF application technique, such as wire loops, arch bars, bone screws or hybrid MMF devices, that yields all advantages and eliminates all disadvantages inherent to each method.

Recent trends favor rapid MMF variants, including Wireless Dental Occlusion Ties (WDOT) for intraoperative use only during simple mandibular fracture repair and hybrid MMF systems for complex fracture patterns requiring rigid intraoperative assembly (monobloc/tension banding) and postoperative guidance into occlusion with elastics (Schopper et al. 2024 [102], Johnson et al. 2024 [103] Akkina et al. 2025 [104]). Based on the literature, it seems the SMARTLock MMF System is better than conventional Erich arch bars (Falci et al. 2015 [72], Edmunds et al. 2019 [27]; Appendix A eContent). The SMARTLock is a viable alternative to EABs in most clinical studies due to advantages in application speed and the use of fewer or no wires. Objective evidence by comparative mechanical stability testing is still pending. Clinical data and laboratory comparison tests are still lacking for the newer members in the league of commercial hybrid MMF devices.

The unique design features of the CHMMFSs impact their handling and functionality. Since tooth root injury remains a possibility with the use of the hybrid devices (Wilt et al. 2019) [29], it is important to consider the targeting function of MMF hardware in terms of the three-dimensional adjustability of the screw-receiving (bone anchor) holes to lessen this risk.

Because of their risk for tooth root damage, MMF screws were long viewed with skepticism. The future will reveal whether the targeting function of hybrid devices will be accepted as a useful parameter for hybrid MMF devices and help overcome any reluctance to their use.

It will also be interesting to see whether the design of anyone of these commercial systems can be more effective in reducing the residual risk of tooth root injury.

## Figures and Tables

**Figure 2 cmtr-18-00033-f002:**
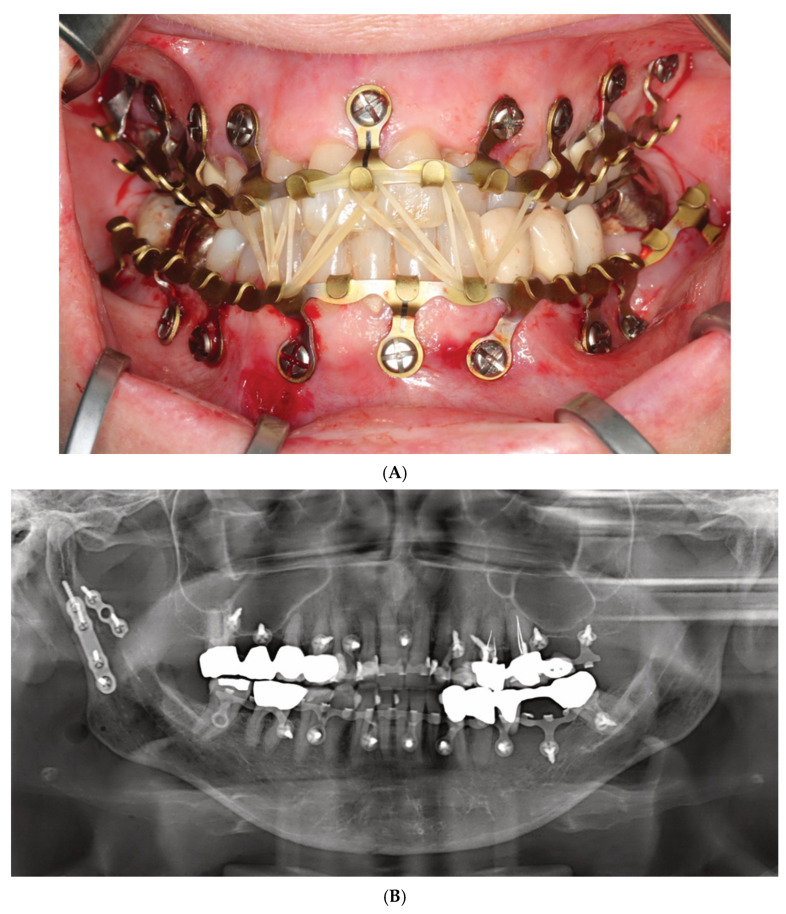
(**A**) Clinical case example—Stryker SMARTLock Hybrid MMF System (regular size) in situ in a right condylar base fracture of the mandible. The arch bars are not shortened; several support legs (‘lugs’) have been bent along the length of their vertical axis to superimpose the screw-receiving holes over the interradicular spaces; all holes except from the right lower 2nd molar are filled with bone screws; the screws are lined up below the mucogingival junction in the upper jaw; in the mandible, the vestibules become shallow along the posterior bucco-alveolar sulcus, so these screws are placed in the mobile mucosa; in the anterior vestibulum, the screws are located low to reach down into the opening interradicular spaces; with such deep placement, the tooth equators cannot be attained. Preinjury occlusion with a lateral crossbite on the left is reestablished and maintained with anterior criss-cross elastic loop intermaxillary fixation. Note the kinking at the base of several bent supporting legs (‘lugs’). (**B**) Immediate postoperative panoramic X-ray after placement of SMARTLock arch bars. Miniplate fixation of right condylar base fracture (ORIF). Interradicular position of all arch bar retaining screws. In the upper right quadrant, 2 screws projecting over maxillary sinus. Eyelet of supporting leg (‘lug’) over right lower 2nd molar empty. Source/origin: Photograph collection—C.P. Cornelius.

**Figure 3 cmtr-18-00033-f003:**
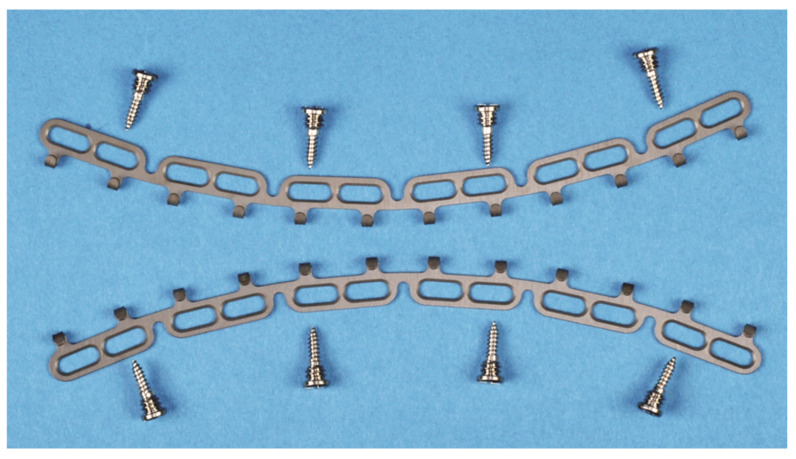
Arrangement of OmniMax arch bars (plates) for application on the mandible and maxillae together with self-drilling locking screws. The plates are preformed by in-plane bending into slight curves. The screws are placed for fixation in the anterolateral transition and the posterior portion of the jaws. At best, no more than two slots in sequence are left empty. Source/origin: Photograph collection—C.P. Cornelius.

**Figure 4 cmtr-18-00033-f004:**
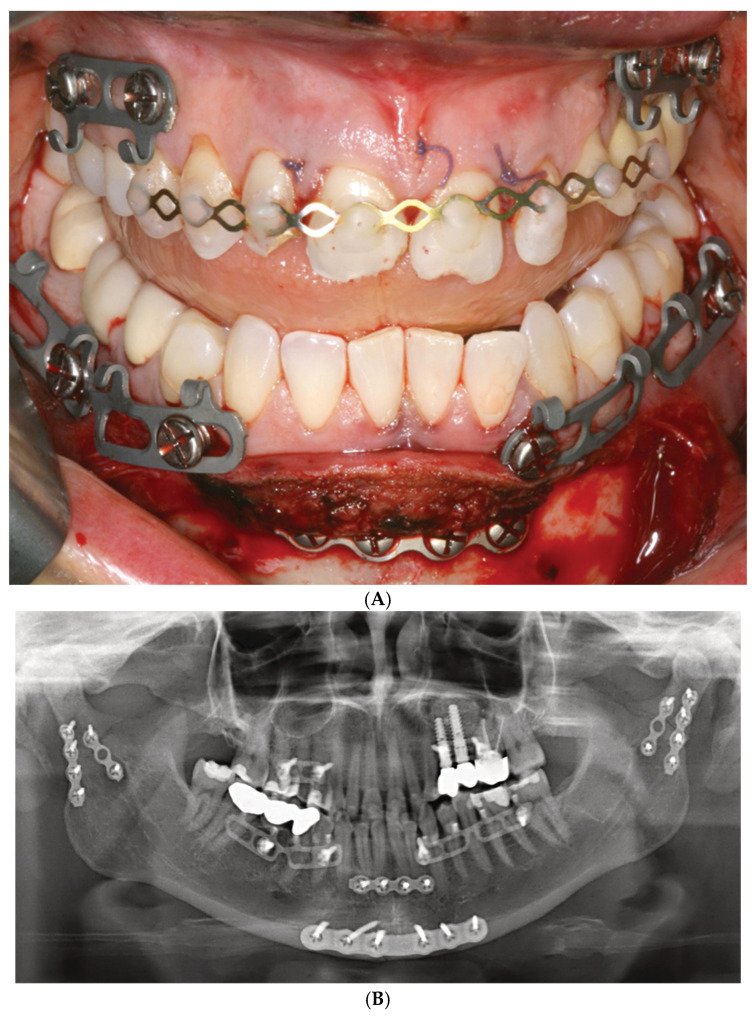
(**A**) Clinical case example—intraoperative view of Zimmer Biomet OmniMax MMF devices divided into 4 segments for treatment of a triple mandibular fracture—bilateral condylar base and symphyseal midline. A dental splint (Titanium Trauma Splint, Medartis, Basel, Switzerland) resin bonded (acid etching technique) to the outer surfaces of the teeth of the anterior maxillary arch supports the repositioned medial upper incisors. The two maxillary segments represent the shortest possible plate variant consisting of a single mounting tab containing two slot apertures, with 2.0 four-hole miniplate visible in open anterior vestibulum approach. Of note: the standoff mechanism has been implemented for all screws—annular screw grooves fully seated in the slots. (**B**) Previous case cont’d. Postoperative panoramic X-ray after placement of OmniMax arch bars, ORIF—via transoral vestibular and preauricular transparotid approaches. Miniplate fixation of the condylar base fractures. Four-hole superior border (tension band) miniplate fixation in combination with a six-hole 2.4 inferior border plate. The arch bar retaining screws appear inserted correctly in the interradicular spaces. Source/origin: Photograph collection—C.P. Cornelius.

**Figure 5 cmtr-18-00033-f005:**
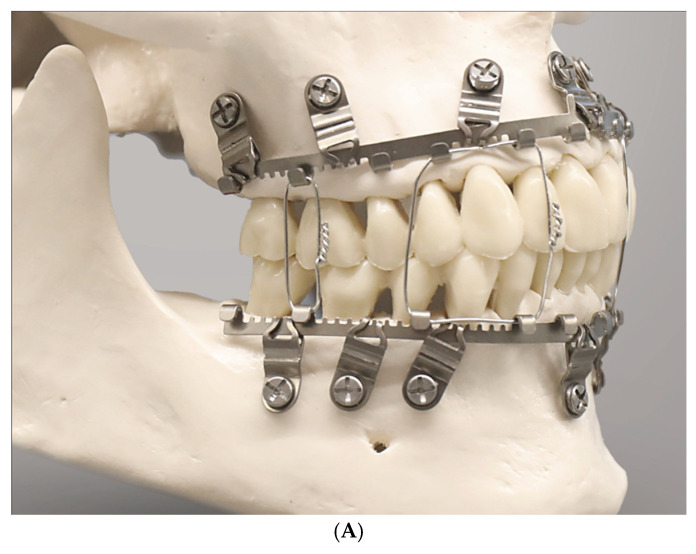
(**A**) Full arch assembly of L1 MMF arch bar/slider plates and intermaxillary wire ligatures in a model—right lateral view. The maxillary bar sections above the incisors and premolars as well as the mandibular bar sections below the lower lateral incisors are void of tabs and gaps (‘teeth’); the angulation of the lateral slider plates and montage under tension maintains the purposed interdigitation between their slots and the arch bar rack. Note: suboptimal asymmetrical vertical placement of lower lateral arch bar sections subsequent to in plane bending alongside the free gingival margins and tooth necks (periodontally unfavorable). (**B**)—frontal view. Angulation of slider plates in upper incisor/canine regions; apart from a midline slider plate the lower incisor and canine region are spared from slider plates accounting for the high risk of tooth root injuries due to narrow interradicular spaces. (**C**)—left lateral view. The slider plate below the lower canine is not properly engaged in the mating bar rack resulting from a too parallel montage—a flaw—in long-term yielding to forces and potentially hazarding stability. Source/origin: L1 MMF System provided by courtesy of KLS Martin, Tuttlingen, Germany—model montage by C.P. Cornelius.

**Figure 6 cmtr-18-00033-f006:**
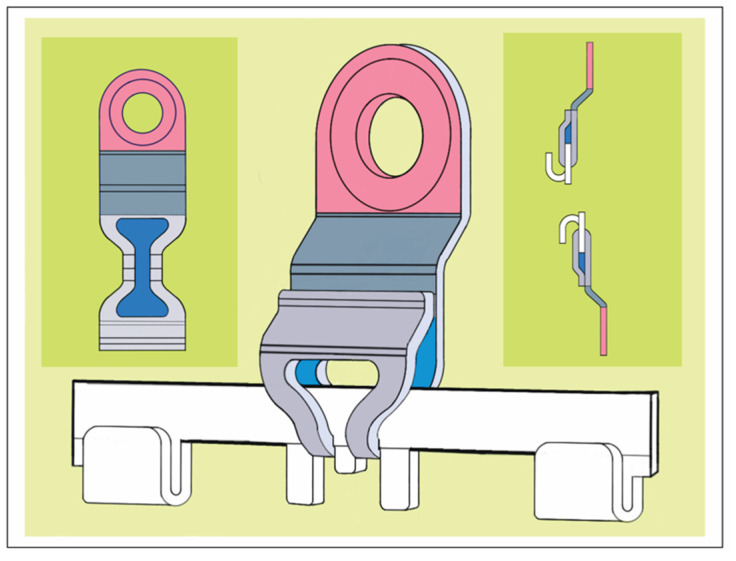
Slider plate snapped in the trident midline halt of the central arch bar section—the shield at the end of the coupling portion abuts the front of the main body like a clip. Interior leeway space between and beyond the extension legs of the coupling portion (blue). (**Inset top left**) Slider plate unfolded to display its three components without overlapping: coupling portion with hour-glass slot design (light grey), main body (dark grey) and screw-receiving hole portion (rouge). (**Inset top right**) Transverse cross sections of mounting plates snapped on arch bars oriented inversely for the mandible and maxillae. Room to move the mounting plates during coupling/removal maneuvers is highly limited antero-posteriorly but open vertically. Source/origin: Modified from 2nd version for the embodiment in Patent No. US 10,470,806 B2—12 November 2019 [12]. Schematic drawing—C.P. Cornelius.

**Figure 7 cmtr-18-00033-f007:**
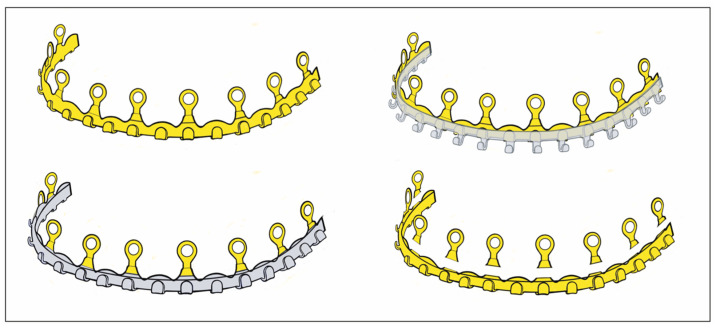
Schemes of a SMARTLock arch bar—oriented and conformed in curvature for maxillary installation—(**top left**) SMARTLock hybrid bar embodiment; (**top right**) superimposition with Erich-type arch bar (note: hook lengths are extended vertically); (**bottom left**) superimposition with conventional Erich arch bar; (**bottom right**) SMARTLock hybrid bar—cutaway illustration subdividing the platform bar and the lugs according to the ‘Erich with lugs’. Source/origin: Schematic drawing—C.P. Cornelius.

**Figure 8 cmtr-18-00033-f008:**
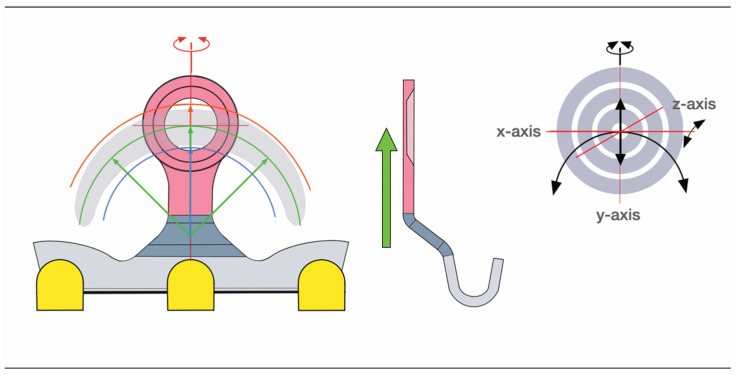
SMARTLock lug/ring opening—maneuverability: (**left**) single lug on a short bar segment—frontal view; (**middle**) single lug, footing and attached bar—lateral profile/cross section. The vertical height of the lug, i.e., the radius or length between the center of the screw hole and middle of the footing (green arrows) and the material properties (thickness, rigidity, hardness, elasticity, elongation at rupture, etc.) predetermine the extent of movements in the frontal plane—precondition: static position of the bar. The radius length can be modified with the angulation of the footing plate. Thus, a crescent surface area gets into reach of the screw-receiving hole. Color coding: footing portion with buckling (dark grey), screw-receiving hole portion (rouge), crescentic surface area (light grey). (**inset—right**) Synoptic scheme: target board with crosshairs and arrows demonstrating possible lug/eyelet movements to locate the screw-receiving (bone anchor) hole into a safe interradicular zone: vertical translation, rotation around longitudinal *z*-axis, anteroposterior (longitudinal/sagittal) translation, minimal rotation around vertical *y*-axis. Source/origin: Schematic drawing—C.P. Cornelius.

**Figure 9 cmtr-18-00033-f009:**
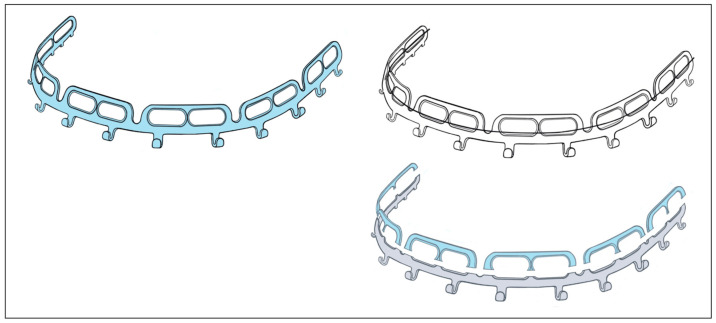
Schemes of the OmniMax embodiment—oriented and conformed in curvature for maxillary application: (**top left**) OmniMax hybrid bar; (**top right**) superimposition with Erich-type arch bar (see Figure 1)—note: hook lengths extended vertically; (**bottom right**) OmniMax hybrid bar—cutaway illustration subdividing the platform bar and the railings. Source/origin: Schematic drawing—C.P. Cornelius.

**Figure 10 cmtr-18-00033-f010:**
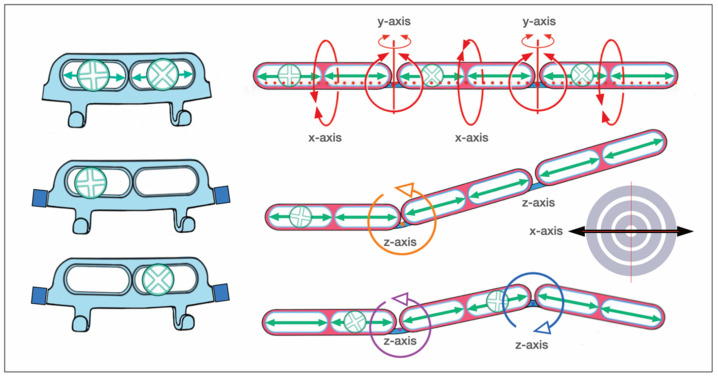
Schemes of the OmniMax arch bar: (**top left**) single bar segment—frontal view, both slots loaded with screws; (**center and bottom left**) bar segment as part of a row, only one slot receives a screw for fixation into an interradicular space; (**top right**) stylized 3-bar segments in a row with 2 pivoting junctions in between—the segments can be hinged along all segments in three spatial axes; (**center right**) a rotational movement around any of the axes—clockwise around the *z*-axis in this example—takes the next (third) segment along into the same direction; (**bottom right**) readjustment of the third segment by a “counter”-clockwise rotation. Color coding: left-right sliding options within a slot (green arrows), screw-receiving slot portions (rouge), pivoting junctions (light blue). (**inset—center far right**) Synoptic scheme: target board with crosshairs demonstrating possible movements inside the slots to locate a safe zone for a screw insertion: transverse/horizontal translation along the *x*-axis. Source/origin: Schematic drawing—C.P. Cornelius.

**Figure 12 cmtr-18-00033-f012:**
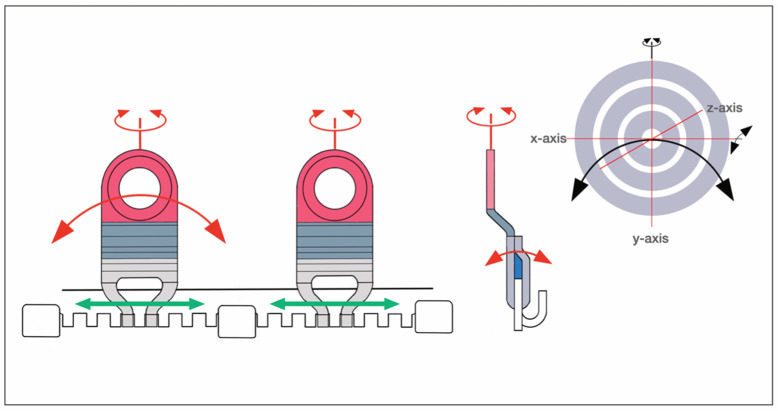
Schemes of the L1 MMF device: (**left**) two segments of a row—frontal view, each engaged with a stackable slider plate—left–right plate moving options within a bar segment (green arrows); (**center**) cross section of a slider plate snapped on a bar unit—leeway space inside the coupling partition (blue) extends vertically. Color coding: slider plate—coupling portion (light grey), main body (dark grey), screw-receiving hole portion (rouge); bar unit with toothed rack segments (white), rotational axes (red). (**inset—right**) Synoptic scheme: target board with crosshairs demonstrating possible movements of the slider plates/screw-receiving holes to locate a safe zone for a screw insertion. Slider plates can be hinged to some degree along all three spatial axes. Source/origin: Schematic drawing—C.P. Cornelius.

**Figure 13 cmtr-18-00033-f013:**
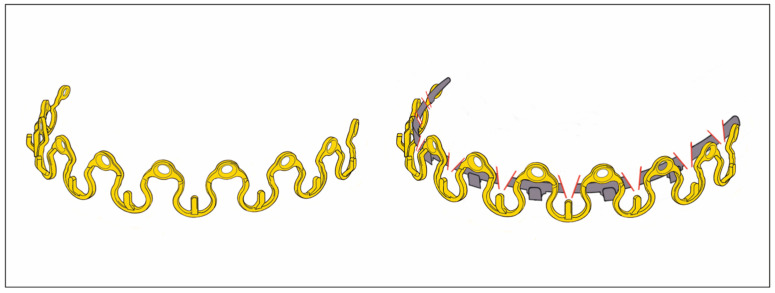
Matrix Wave System embodiment (‘Plate’)—oriented and conformed in curvature for maxillary application—stylized schemes according to the MWP embodiment (Patents No.: US 9,820,77 B2—21 November 2017 [38] and US 10,130,404 B2—20 November 2018 [40]). (**left**) MWP—oriented and conformed in curvature for maxillary application with the bone receiving (bone anchor) holes between crests (on top). (**right**) MWP superimposing an interrupted conventional Erich arch bar. The MWP abandons the band-like Erich arch bar configuration in favor of a periodic sinus wave; in contrast to the serpentine MWP a continuous band structure is capable to embrace the dental arch rigidly and this is frequently put forward as ‘tension band’ function. Source/origin: Schematic drawing—C.P. Cornelius.

**Figure 14 cmtr-18-00033-f014:**
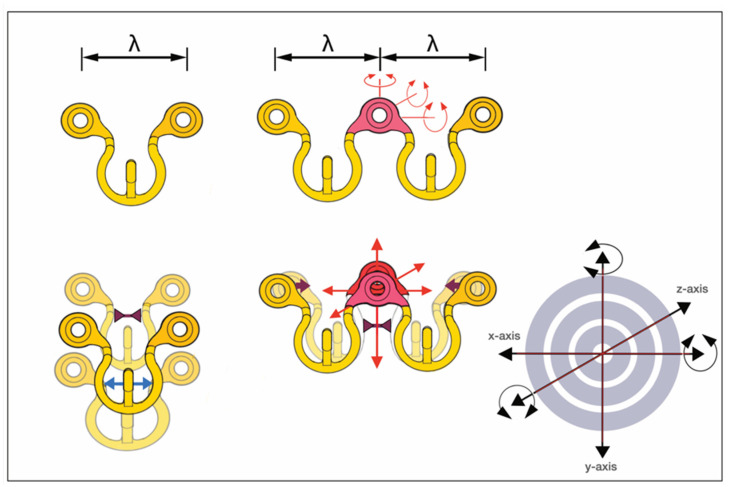
Stylized scheme according to the MWP embodiment (Patents No.: US 9,820,77 B2—21 November 2017 [38] and US 10,130,404 B2—20 November 2018 [40]): (**top left**) one wavelength (λ)/Omega MWP segment can be tailored for omnidirectional placement of screw-receiving holes and/or cleat; (**bottom left**) stretching (blue double arrow) or squeezing (maroon inverted double arrows) in the Omega MWP segment exemplifies just a way to alter its winding, i.e., vertical height in relation to width in one single plane; (**top center**) two-wavelength MWP partition to outline the potential movements of the central screw-receiving hole plateau/flatbed panel (rouge) in all six degrees of freedom—rotational axes are indicated; (**bottom center**) indication of 3-dimensional translational motions; for instance, squeezing of the medial limbs (maroon arrows) increases the vertical height of the plateau with succedent transformation of the wave pattern; (**inset—right**) synoptic scheme: target board with crosshairs demonstrating the potential movements of the mounting plates/screw-receiving holes in six degrees of freedom to target a safe zone for a screw insertion. Source/origin: Schematic drawing—C.P. Cornelius.

**Figure 15 cmtr-18-00033-f015:**
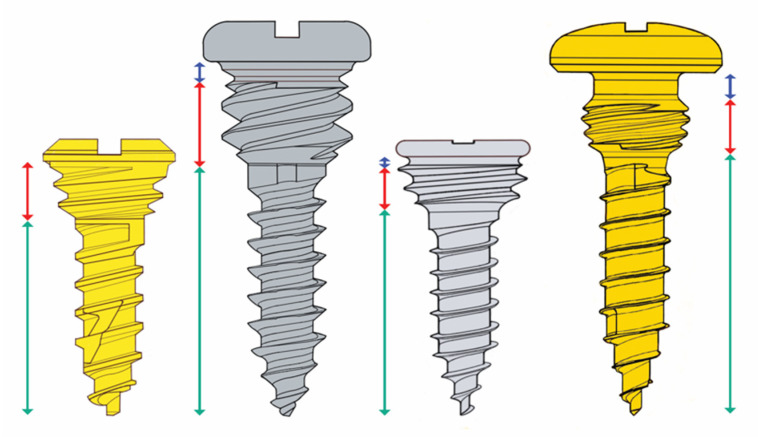
Locking screws of the league of CHMMFSs—frontal views at the same scale. (**From left to right**): Stryker SMARTLock—Stryker (gold); OmniMax—Zimmer Biomet, (silver); L1 MMF—KLS Martin (silver); Matrix Wave—Depuy-Synthes (gold). Double arrows indicate the following: bone insertion threads (green); locking heads/stop drums (red); groove/recess underneath screw heads (blue). The bone insertion threads begin below a neck or transgingival free part. (For more technical characteristics, see text). Source/origin: Schematic drawing—C.P. Cornelius.

**Figure 16 cmtr-18-00033-f016:**
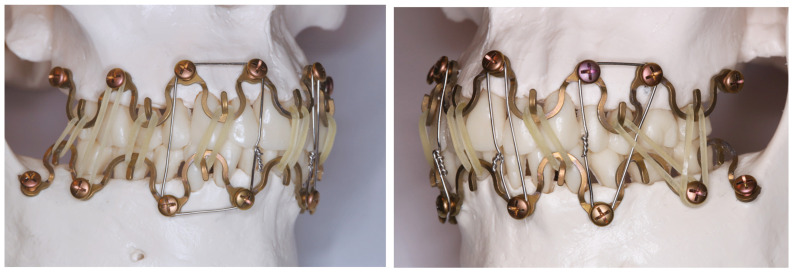
Matrix Wave System full-arch MMF set up—primary intermaxillary fixation with elastic loops across the cleats (hooks/tangs) next to the occlusal plane. Supplementary large span interarch fixation via the prominent heads of the bone fixation screws. (**Left**) Quadrangular wire cerclage cornered by the heads of two pairs of vis-a-vis screws to reinforce the elastic loop intermaxillary fixation. (**Right**) Longitudinal–oval and triangular wire cerclages around two or three opposing screw heads, cleat-to-screw elastic loop connections. Source/origin: Photograph collection—C.P. Cornelius.

**Table 1 cmtr-18-00033-t001:** Synopsis—placement sites for orthodontic miniscrews and likewise for locking screws of hybrid arch bars and commercial hybrid MMF systems.

Maxillae—in the Space Between:
central and lateral incisors
2nd premolar and 1st molar
1st molar and 2nd molar (= intermolar)
**Mandible—in the space between:**
lateral incisor and canine
1st and 2nd premolar
2nd premolar and 1st molar
1st molar and 2nd molar (= intermolar)

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
