# Peer review of "Matrix WaveTM System for Mandibulo-Maxillary Fixation—Just Another Variation on the MMF Theme?—Part II: In Context to Self-Made Hybrid Erich Arch Bars and Commercial Hybrid MMF Systems—Literature Review and Analysis of Design Features"

_1943-3883, 2025, doi:10.3390/cmtr18030033_

Round 1

Reviewer 1 Report

Comments and Suggestions for Authors

Matrix wave system for MMF – just another variation on the MMF theme?

Part I: A review on the provenance, evolution and properties of the system, and

Part II: In context to self-made hybrid Erich arch bars and commercial hybrid MMF systems – Literature review and analysis of design features

I have combined my reviews for both manuscripts as they are intimately related and many comments relate to both, plus they both have the same authors

Initially and then on review, I was somewhat overwhelmed by the size of each article, at 42 and 67 pages respectively, with a further 16 pages of supplement, amounting to 125 pages in total, this seemed to be more a chapter of or even a textbook in its own right

The overall concept and design of both papers is interesting and informative, although they do not necessarily present any new information, there is deep analysis of the design of several systems (esp. the Matrix Wave Plate – MWP - in Part I) and more than enough information on which the reader can make an informed choice on which system to choose, when and for what

Further, there is enough information, and some, to analyse the performance, the pros and cons of each system with reference to the available literature. The limitations, variations, inconsistencies and omissions of many relevant studies are compared and contrasted/analysed in detail

In particular, the section on dividing and sectioning pieces or sections of the MWP to use in different parts of each arch was extremely useful and instructional. Further, the introduction, discussion and conclusion sections of each part are also well written and contain the essential messages for readers to take away

So, in summary, these are useful and helpful in the context of the available literature, however, crystal clear messages are difficult to obtain from the main body of each manuscript and there are many points that must be considered in terms of critique:

  1. Parts I and II could be significantly condensed into 1 whole article entitled: A comparative literature review on variations of MMF systems, with specific focus on the Matrix Wave System – this will improve accessibility for readers and not discourage them with 2 large parts, a supplement and 2 long titles
  2. I suggest that parts I and II are reversed so that the comparison of the available systems is presented 1st and then the 2nd part concentrates on the when, how and why of the MWP system and where it may be superior to its MMF counterparts
  3. A thorough and forensic English language, grammar and tense review/proof read is required to make these manuscripts more concise and remove Americanised English plus improve sentence structure; many sections in both papers are too long, overly detailed, repetitive and verbose. In their current form, I have no doubt that readers will be disincentivised and demotivated by the amount of detailed technical language in each paper. The authors even allude to this point in their own conclusion at the end of the current part II (lines 1937-40). It is asking an enormous amount of readers to expect them to concentrate on these articles alone for over 100 pages
  4. Simple diagrams of each system are all that are required, with small technical descriptions. If the authors wish to include system minutiae/technical differences/specifications and theoretical geometric representations, then they can, but they need to be in electronic supplement/appendix form, which readers can reference/read if they wish – I suggest this is done via a QR link to the CMTR website to avoid a large paper volume
  5. There is no detailed mention of the use of custom made arch bars from internal laboratories and yet these are still in use. These can have longer, slender cleats that are more user friendly than the small cleats on EABs or hybrid versions thereof and may result in a different set of comparative data. The MWP cleat design is very similar to the custom made arch bar type of cleat, and there is no doubt in practice that it is easier to place wire and elastics around MWP cleats than the cleats of the other commercially available EAB/Hybrid types, further, the smooth screw head caps can also be used as an extra bonus
  6. There is no mention of the risk of and complication of MMF screw (alone or in combination with hybrid arch bars) head fractures, which are known/experienced and can result in retained screw threads
  7. There is no mention of the type of screwdriver used for MMF devices – be that a press/friction fit or a sleeve type screwdriver – this is relevant because in clinical practice, although slightly bulky the sleeve type results in less disconnection of the screw especially in difficult to access areas in the oral cavity
  8. Although tooth/teeth injury as a consequence of/risk of screw placement is mentioned repeatedly, there is no evidence presented on the need for, or incidence of, root canal treatment/extractions as a consequence of these iatrogenic injuries, plus they indicate that they will often go on to heal. Neither is it indicated if the teeth injuries were different between different experience levels of operator
  9. If there is a concern about teeth injuries, then why haven’t custom made hybrid arch bars become part of routine maxillofacial trauma practice? So that screw positions can be guided from the outset to avoid roots? Especially, as were now in the age of 3D/computer planning, with widespread availability and access to CT scans and planning software
  10. Even though wire stick injury is a risk and is obviously important, there is no evidence presented in terms of the risk/incidence of Blood Born Virus infection (BBV) as a direct consequence of wire stick injury. Indeed, it could be theorised that as wires are solid (not hollow like needles), that the risk of transmission of BBV and tissue should be extremely low. This point refers to part II line 1363
  11. In terms of very specific corrections that are required:

  1. Through part I and II the following words would benefit from change, to increase access for a broad readership:

anatomical (not anatomic), anaesthesia (not anaesthesia), trauma (not traumatization), emphasised (not emphasized), mouldable (not moldable), bridle (not bridal), bevelled (not bevelled), block (not bloc +/-mono), calorific (not caloric), favouring (not favoring), minimise (not minimize), immobilisation (not immobilization), specialised (not specialized), summarised (not summarized), characterise (not characterize), labelled (not labelled), analyse (not analyse), scrutinised (not scrutinized), devitalisation (not devitalization)

  1. Specific line corrections suggested by line are:

The current Part I:

145: oscillating, not oscillation

193: remove on after minimise

311: change what to so

348: remove of course

413-16: has there ever been a comparison of single MWP segment cost/performance to single IMF screw use?

460: replace what with which

724: self-cutting screws are more commonly termed self-drilling

742: change establishing: to namely:

760: start with Due to suspected study bias

762: change to as follows at the end of the sentence

769: add use after screw

780: change to and slow application, at end

794: change incidents to incidence

References: 1,2,7,8,12,13,14,18,19,5,58,74 – have all of these been translated into English for all authors to read/use and refer to? and are translated versions available for readers should they wish to access them?

To correct a change in format to references 37,39

What is a FAMI screw in reference 57??

Lastly, can a reference to the submitted but not yet accepted part II, be used in part I?

The current part II:

Introduction: can a reference to the submitted but not yet accepted part I, be used in part II?

This point also applies to lines 902-4, 1113, 1726

142-3: change likewise to like

174: place a gap between devices and in

Why are there separate supplements, if all of the text from these is already included in the body of the article? I suggest one or the other

234: change countersunk to countersink

261: change to lean to leaning of

270: change embodiment to arch bar

Page 13 onwards:

24-5: is 50% perforation referring to length of screw in root or width of root involved?

36: did any root perforations result in Root Canal Treatment or extraction? Plus 1594

81: change in succession of to after

154: change mandible to mandibular

239: remove 0 after indications

277: change remainder to remaining

297: change allowing to allow

359: add of after application

391: change resumed to reviewed

449: remove that, after so

450: change withdraw to withdrew, add as it after ambiguity, change repeating to repeated

451: change leaving to left

453: change past-time to previous

538: change of to the

548: change damages to damage sites

559: change for to of

564: change need to needed

565: change gingival to gingivae, remove of before compromised

573: change of, at end, to for

608: what is MFF? Is it MMF?

614: remove a

629: change consonant to consistent or coincident

646: change in- to incisions

659: add is after system and….

672: change has to have

762: what is OP? is it operative?

908-11: all text part of same sentence and no need for capital letters on patients and detail

1054:  change bend to bent

1145: change does to do

1146: put which after threads

1182: change Walter to Wolter

1201: add leading after plate

1202: change the start of the sentence to with the, add may after height

1216: change “recommendable to” to “recommend that” and add is obtained, after situations

1349: change fractures to fracture

1402: add “due” before “to clamping”

1415: change inserting to insertion

1435: change what to which, add that it can also take them out of reach of cost restrained public funded health care systems

1447: change assumingly is, to is assumed to be:

1451: add general before anesthesia and change anesthesia to anaesthesia

1463: change as to like

1464: change as to like

1482: put space between demographic and variables

1517: put and after thick

1536: is it correct to use “long-term” when screws in this context are short-term use?

1544: should not simply be relied upon needs to be moved to the end of the sentence

1553: change cement to cementum

1554: change dentin to dentine

1560: change damages to damage

1627: change to heterogeneous

1670: change conceptionally to in concept

1702: change to had to rule out

1714: change assumed to used

1753: change connotes to refers to

1793: change to hardware extensions and MMF hybrid type screw heads is evidenced in part, but is both technique and hardware dependent

1836: move hints to after unambiguous

1853-54: could the differences in patient outcomes also be attributed to the use of wire or elastic IMF? As there will be different amounts of possible jaw movement and opening with each

1887-88: might this be explained by the use of the hand-held fracture reduction technique?

1917: change resonance to, to, acceptance of

1935: change to biased, and to placed after the word inattention

1939: remove resulting

1946: change to on after impact

1953: remove a after establishing

1954: put of after component

1955: change solutions to devices

1959: put simple after during

1969: put on after impacts

References: 11,12,13,14,15,16,44,46,48 – all have their format changed and just needs to be made the same as the other references

Has reference 55 been translated like in part I?

2326: is it meant to be intentional or unintentional damage with mini screw implants in this reference?

Reference 115: can this be used as a reference if the article is still in preparation and not accepted yet?

Comments on the Quality of English Language

As above

Reviewer 2 Report

Comments and Suggestions for Authors

I have been asked to review Part 1 and part 2 of this subject. This review covers both manuscripts.

I can see the amount of hard work that has gone into developing these two manuscripts with Part 1  discussing  the new Omega shaped Wave Plate and its origins and part 2 looking at the history of Intermaxillary Fixation.  It is an important and relevant subject. I appreciate there is no word count for the journal, but I feel both manuscripts are too long to keep the reader interested. Part 2 is 67 pages!  You begin to loose site of what the take home message is.

Looking at the radiographic images I’m not sure what the purpose of the IMF is in the post operative period? In Part 1  with Figure 16  and part 2  Figures 2,  4  & 5 all are post op  x rays where all  fracture sites have be reduced and fixated with rigid plates yet the IMF is still present.   What benefit does the IMF have post operatively when rigid plates are in place?

The discussion on MWF fixation  is repeated again in Part 2 when it’s already been discussed in part 1

I feel that the same messages can be made with half the word content in both parts  and a significant of images can be removed too.

Round 2

Reviewer 2 Report

Comments and Suggestions for Authors

The manuscript has been curtailed accordingly. However, I'm still concerned about the use of IMF when rigid plating has been applied. A comment needs to be made regarding this
